# Treatment Effects Estimation By Uniform Transformer

**Ruoqi Yu**
University of Illinois Urbana-Champaign
Champaign, IL 61820, USA
`ruoqi.yu.ry@gmail.com`

**Shulei Wang**
University of Illinois Urbana-Champaign
Champaign, IL 61820, USA
`shuleiw@illinois.edu`

## Abstract

In observational studies, balancing covariates in different treatment groups is essential to estimate treatment effects. One of the most commonly used methods for such purposes is weighting. The performance of this class of methods usually depends on strong regularity conditions for the underlying model, which might not hold in practice. In this paper, we investigate weighting methods from a functional estimation perspective and argue that the weights needed for covariate balancing could differ from those needed for treatment effects estimation under low regularity conditions. Motivated by this observation, we introduce a new framework of weighting that directly targets the treatment effects estimation. Unlike existing methods, the resulting estimator for a treatment effect under this new framework is a simple kernel-based $U$-statistic after applying a data-driven transformation to the observed covariates. We characterize the theoretical properties of the new estimators of treatment effects under a nonparametric setting and show that they are able to work robustly under low regularity conditions. The new framework is also applied to several numerical examples to demonstrate its practical merits.

## 1 Introduction

In order to infer causal relations in an observational study, a major difficulty is to reduce the bias brought by the confounding covariates related to both the treatment assignment and the outcome of interest (Imbens & Rubin, 2015). This task can be accomplished by balancing the empirical distributions of observed covariates in different treatment groups. One common strategy to adjust imbalances of confounders is weighting (Rosenbaum, 1987; Robins et al., 1994; 2000; Hirano & Imbens, 2001; Hirano et al., 2003), which seeks a weight for each sample so that covariates distributions are similar between the weighted groups.

A conventional approach widely employed in the literature for estimating weights is the inverse-probability weighting (IPW) method, where the weight of each sample is the corresponding inverse probability of receiving the treatment (Horvitz & Thompson, 1952; Hahn, 1998; Robins et al., 2000; Hirano & Imbens, 2001; Hirano et al., 2003). It has been shown that this method can entirely remove the bias in the estimation of treatment effects when the true propensity score, defined as the conditional probability of receiving treatment given the covariates (Rosenbaum & Rubin, 1983), is used. Practical applications of IPW usually require estimating the propensity score based on a presumed model, as the true propensity score is typically unknown in advance. However, misspecification of the propensity score model can induce large biases in estimating treatment effects using IPW (Kang & Schafer, 2007). This observation motivates recent works to develop more robust weighting methods, aiming to estimate the propensity score or the weight itself by directly comparing some prespecified moments/basis functions of covariates between the treatment groups (Graham et al., 2012; Hainmueller, 2012; Imai & Ratkovic, 2014; Zubizarreta, 2015; Chan et al., 2016; Zhao & Percival, 2016; Wong & Chan, 2018; Zhao, 2019; Wang & Zubizarreta, 2020; Kallus, 2020; Hirshberg & Wager, 2021; Singh, 2021; Bruns-Smith et al., 2023; Fan et al., 2023). These direct balancing weighting methods have been shown to work more robustly than the IPW method in practice.

The good performance of the inverse-probability weighting or direct balancing weighting methods usually relies on strong regularity conditions for either the propensity score or the response func-

tions. Specifically, the smoothness levels of the propensity score and the response functions must be at least as large as half of the covariate dimension in these methods (Chan et al., 2016; Fan et al., 2023; Wong & Chan, 2018; Wang & Zubizarreta, 2020). However, it is not immediately clear how to construct the weights when these criteria are not met and to what extent the treatment effects could be estimated by the weighting method in such non-smooth cases. Therefore, this paper aims to address these issues and develop a new framework of weighting methods to fill these needs.

We first investigate the weighting methods from a functional estimation perspective as a treatment effect is essentially a functional of the response and density functions. Through this perspective, we argue that the best way to estimate the ideal weights does not necessarily lead to the most efficient weighting estimator for treatment effects in the non-smooth case. In other words, the weights needed for covariate balancing could be different from the weights needed to estimate treatment effects when the response and density functions are non-smooth. Then, a natural question arises: how can we design a weighting method that directly targets the treatment effects estimation?

To address this question, we introduce a new weighting framework called Weighting by Uniform Transformer (WUNT). Note that the term "transformer" here differs from its conventional usage as neural network architecture (Vaswani et al., 2017). Our motivation for this new framework stems from a noteworthy insight into the uniform transformer, defined as a transformation mapping the covariate distribution in the control group to a uniform distribution. The uniform transformer gives us a clean form of the weighting method, allowing us to directly make an accurate trade-off between the bias and variance for the treatment effects estimation. In addition, we show how to construct a data-driven uniform transformer from the covariates of control samples in a computationally efficient way. With the data-driven uniform transformer, the weights in WUNT are customized for the treatment effects estimation, and the resulting weighting estimator is a kernel-based $U$-statistic.

To demonstrate the merits of the newly proposed framework WUNT and the corresponding estimators , we study the theoretical properties under a nonparametric setting, especially when the response and density functions are non-smooth. Specifically, we show that the proposed estimator is consistent under very mild conditions. In addition, if the covariate density in the control group is known or can be estimated accurately, the minimax optimal converge rate for the mean square error of estimating the average treatment effect on the treated group is

$$n^{-\frac{4(\alpha+\beta)}{d+2(\alpha+\beta)}} + n^{-1},$$

where $n$ is the sample size, $d$ is the dimension of covariates, and $\alpha$ and $\beta$ are the smoothness levels of the response surfaces and density functions of the covariates, respectively. This result suggests that estimation of the treatment effects becomes more difficult when the response and density functions are less smooth. The converge rate presented here also appears in Robins et al. (2008; 2009; 2017), where an accurate trade-off between bias and variance is achieved by semi-parametric methods with additional bias reduction techniques. Our result shows that the newly proposed weighting method is also able to do so because of the uniform transformer. To our best knowledge, this is the first minimax rate-optimal weighting method when the response and density functions are non-smooth. Practical merits of WUNT are further demonstrated through simulation experiments in Appendix B.

## 2 PROBLEM SETTING AND WEIGHTING METHODS

### 2.1 PROBLEM SETTING AND NOTATIONS

Suppose that the observed data $(\vec{X}_i, Z_i, Y_i)$, $i = 1, \ldots, n$ are independent and identically distributed observations of $(\vec{X}, Z, Y)$, where $\vec{X} \in \mathbb{R}^d$ are the observed covariates, $Z$ is a binary indicator variable for the treatment and $Y$ is the outcome of interest. Under the potential outcome framework for causal inference (Rubin, 1974; Imbens & Rubin, 2015), $Y^0$ and $Y^1$ are the potential outcomes if the individual is assigned to the treated ($Z = 1$) or control group ($Z = 0$). Then, the observed outcome can be written as $Y = (1 - Z)Y^0 + ZY^1$. Throughout this paper, we always assume the strong ignorability of the treatment assignment (Rosenbaum & Rubin, 1983)

$$\{Y^0, Y^1\} \perp Z \mid \vec{X} \qquad \text{and} \qquad 0 < \mathbb{P}(Z = 1 | \vec{X}) < 1. \tag{1}$$

It is of interest to estimate the average treatment effect (ATE), $\tau_{\text{ATE}}$, or the average treatment effect on the treated group (ATT), $\tau_{\text{ATT}}$. Let $\mu_T(\vec{X}) = \mathbb{E}(Y^1 | \vec{X})$ and $\mu_C(\vec{X}) = \mathbb{E}(Y^0 | \vec{X})$, we have

$$\tau_{\text{ATE}} = \mathbb{E}(\mu_T(\vec{X}) - \mu_C(\vec{X})) \qquad \text{and} \qquad \tau_{\text{ATT}} = \mathbb{E}(\mu_T(\vec{X}) - \mu_C(\vec{X}) | Z = 1).$$

For the sake of concreteness, we focus primarily on the average treatment effect on the treated group $\tau_{\text{ATT}}$ in this paper. The techniques are also applicable to more generalized cases, e.g., Section C.1 of the Appendix discusses the robust estimation for the average treatment effect $\tau_{\text{ATE}}$ under the new framework. Let $f_T(\vec{X}) = \mathbb{P}(\vec{X}|Z = 1)$ and $f_C(\vec{X}) = \mathbb{P}(\vec{X}|Z = 0)$, then $\tau_{\text{ATT}}$ can be written as

$$\tau_{\text{ATT}} = \mu_{TT} - \mu_{CT} = \int \mu_T(\vec{X})f_T(\vec{X})d\vec{X} - \int \mu_C(\vec{X})f_T(\vec{X})d\vec{X}.$$

It is natural to estimate the first term $\mu_{TT}$ by $\sum_{i=1}^{n} Y_i Z_i / \sum_{i=1}^{n} Z_i$. The second term $\mu_{CT}$ is the major challenge in estimating $\tau_{\text{ATT}}$ since the data with both response function $\mu_C(\vec{X})$ and sampling distribution $f_T(\vec{X})$ are inaccessible. Therefore, the main parameter of interest in this paper is $\mu_{CT}$.

## 2.2 Weighting Methods: A Functional Estimation Perspective

In general, a weighting method (Rosenbaum, 1987; Hirano et al., 2003) seeks weights for each control sample so that covariates of the weighted control samples are more similar to those of the treated samples. Given the weights $w_i$ for each sample, $\mu_{CT}$ is estimated by the weighted mean

$$\hat{\mu}_{CT} = \sum_{i=1}^{n} w_i (1 - Z_i) Y_i. \tag{2}$$

In this section, we discuss the weighting methods from a functional estimation perspective as $\mu_{CT} = \int \mu_C(\vec{X})f_T(\vec{X})d\vec{X}$ is essentially a bilinear functional of the response and density functions. The main intuition behind weighting is that the functional $\mu_{CT}$ can be rewritten as

$$\mu_{CT} = \int \mu_C(\vec{X})\frac{f_T(\vec{X})}{f_C(\vec{X})}f_C(\vec{X})d\vec{X} = \int \mu_C(\vec{X})f_C(\vec{X})w(\vec{X})d\vec{X}, \tag{3}$$

where the weighting function $w(\vec{X})$ is

$$w(\vec{X}) = \frac{f_T(\vec{X})}{f_C(\vec{X})} = \frac{\pi(\vec{X})}{1 - \pi(\vec{X})}\frac{\mathbb{P}(Z = 0)}{\mathbb{P}(Z = 1)}.$$

Here, the weighting function at each $\vec{X}_i$ can be treated as ideal weights for the weighting methods since $w(\vec{X})$ can make the distributions in the treated and control groups perfectly balanced. In practice, the weighting methods aim to estimate the weighting function at each $\vec{X}_i$ by some estimator $\hat{w}(\vec{X}_i)$ and then replace $w(\vec{X})$ by $\hat{w}(\vec{X})$ in equation 3. Despite the difference between IPW and direct balancing weighting, the common goal is to estimate the ideal weights $w(\vec{X})$.

On the other hand, the ultimate goal in treatment effects estimation is to estimate $\mu_{CT}$ rather than $w(\vec{X})$, so the weighting methods focusing on estimating the ideal weights can be seen as a plug-in estimator for $\mu_{CT}$ since $w(\vec{X})$ is replaced by $\hat{w}(\vec{X})$ in equation 3. However, such a plug-in strategy does not necessarily lead to an efficient estimator for the functional $\mu_{CT}$ (Lepski et al., 1999; Newey et al., 2004; Cai & Low, 2011; Robins et al., 2017) because the best estimator for $w(\vec{X})$ might not be the most suitable choice for estimating $\mu_{CT}$. An explicit example is given to illustrate this point in Section 2.3. Therefore, a natural question arises: can we design the weights in equation 2 aiming at estimating $\mu_{CT}$ directly? In this paper, we demonstrate that this is indeed possible; in fact, the weights needed for estimating $\mu_{CT}$ are over-debiased and variance-inflated estimators for $w(\vec{X})$ when the response function in the control group is not smooth.

## 2.3 A Warm-Up Example

To illustrate the idea, we start with a special case where the covariate $\vec{X}$ is one-dimensional ($d = 1$) and the distribution of $\vec{X}$ in the control group $f_C(\vec{X})$ is the uniform distribution on $[0, 1]$. For simplicity, we assume $f_T(\vec{X}) = 0$ at $\vec{X} = 0$ and 1 to avoid boundary bias. In this case, the weighting function $w(\vec{X})$ becomes $f_T(\vec{X})$, so estimating the weights is equivalent to estimating the density of

covariates in the treated group. To estimate the density $f_T$ without assuming any parametric form, consider using one of the most commonly used density estimators – the kernel density estimator

$$\hat{f}_T(x) = \frac{1}{n_1 h} \sum_{i=1}^{n} K\left(\frac{x - \vec{X}_i}{h}\right) Z_i = \frac{1}{n_1} \sum_{i=1}^{n} K_h\left(x - \vec{X}_i\right) Z_i,$$

where $K(\cdot)$ is a one-dimensional kernel function and $h$ is the bandwidth. Standard analysis for kernel density estimators suggests that

$$\mathbb{E}(\hat{f}_T(x) - f_T(x))^2 \lesssim \underbrace{h^{2\beta}}_{Bias} + \underbrace{1/(nh)}_{Variance}$$

if we assume $f_T(x)$ belongs to Hölder class $\mathcal{H}^\beta([0,1])$; the formal definition of Hölder class is introduced in Section 4. See Tsybakov (2008) for a detailed proof. Therefore, if we aim to estimate the weighting function $w(\vec{X})$, the bandwidth $h$ should be chosen as $h \asymp n^{-1/(1+2\beta)}$, i.e., $h$ is of the same order as $n^{-1/(1+2\beta)}$. Is this bandwidth also the most suitable one for estimating $\mu_{CT}$?

It seems reasonable to expect that the best way to estimate the weighting function $w(\vec{X})$ leads naturally to the best weighting estimator for $\mu_{CT}$ and hence the ATT. However, we now show that the bias and variance trade-off for estimating the weighting function $w(\vec{X})$ can be very different from that for estimating $\mu_{CT}$. If the weights in equation 2 are replaced by the above kernel density estimator, the resulting weighting estimator for $\mu_{CT}$ is then

$$\hat{\mu}_{CT} = \frac{\sum_{i_1,i_2=1}^{n} Y_{i_1}(1 - Z_{i_1}) K_h\left(\vec{X}_{i_1} - \vec{X}_{i_2}\right) Z_{i_2}}{\sum_{i_1,i_2=1}^{n} (1 - Z_{i_1}) K_h\left(\vec{X}_{i_1} - \vec{X}_{i_2}\right) Z_{i_2}}.$$

Since $\hat{\mu}_{CT}$ is a $U$-statistics, our analysis in Section 4 shows that

$$\mathbb{E}(\hat{\mu}_{CT} - \mu_{CT})^2 \lesssim \underbrace{h^{2(\alpha+\beta)}}_{Bias} + \underbrace{1/n + 1/(n^2 h)}_{Variance},$$

if we further assume the response surface in the control group $\mu_C(x)$ belongs to Hölder class $\mathcal{H}^\alpha([0,1])$. The proof is omitted here since the result is a special case of Theorem 1. Unlike estimating the weighting function $w(\vec{X})$, the bias in estimating $\mu_{CT}$ also relies on the smoothness of the response function. The new bias and variance trade-off suggests that the optimal choice of the bandwidth $h$ for estimating $\mu_{CT}$ is

$$h \asymp \begin{cases} n^{-2/(1+2(\alpha+\beta))}, & \alpha + \beta \leq \frac{1}{2} \\ [n^{-1}, n^{-1/2(\alpha+\beta)}], & \alpha + \beta > \frac{1}{2} \end{cases},$$

where $h \asymp [a,b]$ means $h \asymp c$ for arbitrary $c \in [a,b]$. When the response function in the control group $\mu_C(x)$ is smooth enough, i.e., $\alpha \geq 1/2$, the optimal bandwidth for estimating the weighting function is also optimal for estimating $\mu_{CT}$. This explains why we can estimate the ATT efficiently through targeting the covariate balancing in this case. On the other hand, if the response function $\mu_C(x)$ is non-smooth, i.e., $\alpha < 1/2$, the optimal bandwidth for estimating $\mu_{CT}$ is much smaller than the optimal choice for estimating $w(\vec{X})$. In particular, we need to take the smoothness of the response function into account when we choose the optimal bandwidth for estimating $\mu_{CT}$. Putting differently, the optimal choice of $h$ for estimating $\mu_{CT}$ can result in a suboptimal estimator for $w(\vec{X})$, which is over-debiased and variance-inflated. The similar phenomenon is also observed in estimation of integrated squared density (Bickel & Ritov, 1988; Laurent, 1996; Giné & Nickl, 2008). This example not only illustrates that the optimal estimator for $w(\vec{X})$ does not necessarily lead to an efficient estimator for $\mu_{CT}$ but also suggests a potential strategy to design the weights tailored for treatment effects estimation.

## 3 WEIGHTING BY UNIFORM TRANSFORMER

### 3.1 A WEIGHTING FRAMEWORK FOR TREATMENT EFFECTS ESTIMATION

The warm-up example above suggests that the weights targeting the weighting function $w(\vec{X})$ may not lead to an efficient estimator for $\mu_{CT}$ and hence the ATT. In this section, we build upon these insights and introduce a new weighting framework tailored for treatment effects estimation.

A unique feature of the warm-up example in Section 2.3 is that the covariate follows a uniform distribution in the control group. Because of this property, the weights estimation problem is reduced to a density estimation problem. To apply this technique to covariates with any distribution, we can map the covariate distribution in the control group to a uniform distribution. Any such transformation $\Phi$ is referred to as a "uniform transformer" in this paper. We assume the uniform transformer is known in the current section and leave the discussion on the construction of a uniform transformer to the next section. Let $\vec{U} = \Phi(\vec{X})$ denote the data after transformation. Since $f_C^\Phi(\vec{U})$ is a uniform distribution density, we can rewrite the propensity score-based weight $w_i$ as

$$w_i \propto \frac{\pi(\vec{X}_i)}{1 - \pi(\vec{X}_i)} \propto \frac{f_T(\vec{X}_i)}{f_C(\vec{X}_i)} \propto \frac{f_T^\Phi(\vec{U}_i)}{f_C^\Phi(\vec{U}_i)} \propto f_T^\Phi(\vec{U}_i).$$

That is, to estimate the weights $w_i$, we only need to estimate the density $f_T^\Phi(\vec{U})$ at each $\vec{U}_i$. However, as demonstrated in the warm-up example, the density estimation should be done carefully since the optimal tuning parameters targeting the density estimation itself can differ from the optimal choices for estimating the treatment effects. In other words, the key difference between our framework and the classical density estimation problems is the choice of tuning parameters. Under this new framework, the weights in equation 2 can be constructed in two steps: (i) transform the covariates $\vec{X}_i$ by a uniform transformer $\Phi$, (ii) estimate the weights by any density estimator with tuning parameters targeting the treatment effects estimation. The framework is summarized in Algorithm 1, and we call it "weighting by uniform transformer" (WUNT).

---

**Data:** $\{(\vec{X}_i, Z_i, Y_i)\}_{i=1}^n$.
**Result:** Weights $w_i$ and an estimator of $\mu_{CT}$.
Construct the uniform transformer by $\{\vec{X}_i\}_{i:Z_i=0}$ and apply transformation for all data
$\vec{U}_i = \Phi(\vec{X}_i)$;
Estimate $\hat{f}_T^\Phi(\vec{U})$ from $\{\vec{U}_i\}_{i:Z_i=1}$;
Evaluate the weights by $w_i = \hat{f}_T^\Phi(\vec{U}_i)/\sum_{i:Z_i=0} \hat{f}_T^\Phi(\vec{U}_i)$ for $Z_i = 0$;
Assign the weights $w_i = 1/n_1$ for $Z_i = 1$;
Estimate $\mu_{CT}$ by equation 2;

---

**Algorithm 1**. Weighting by Uniform Transformer (WUNT)

For the density estimation in the second step, we introduce two of the most popular nonparametric density estimators in the literature. The first density estimator is the kernel density estimator, which has been widely used in many applications. The kernel density estimator is defined as

$$\hat{f}_T^\Phi(\vec{U}) = \frac{1}{n_1} \sum_{i:Z_i=1} \frac{1}{\det(H)} K\left(H^{-1}(\vec{U} - \vec{U}_i)\right) = \frac{1}{n_1} \sum_{i:Z_i=1} K_H\left(\vec{U} - \vec{U}_i\right),$$

where $K(\cdot)$ is a kernel function and $H = \mathrm{diag}(h_1, \ldots, h_d)$ is a diagonal matrix of the bandwidths which controls the amount of smoothing. Let $K_H$ denote the scaled kernel with bandwidth matrix $H$ and write $K_H$ as $K_h$ when $h_1 = \ldots = h_d = h$. In particular, we assume $K(\vec{X}) = G(X_{(1)}) \times \ldots \times G(X_{(d)})$, where $G(\cdot)$ is a univariate kernel $\int G(x)dx = 1$. We call kernel $K$ an $\alpha$ order kernel if $\int x^t G(x)dx = 0$ for any integer $t \leq \alpha$ and $\int |x^\alpha G(x)|dx < \infty$. With the kernel density estimator, the final estimator of $\mu_{CT}$ in Algorithm 1 can be written as

$$\hat{\mu}_{CT} = \frac{\sum_{i_1,i_2=1}^n Y_{i_1}(1 - Z_{i_1})K_H(\Phi(\vec{X}_{i_1}) - \Phi(\vec{X}_{i_2}))Z_{i_2}}{\sum_{i_1,i_2=1}^n (1 - Z_{i_1})K_H(\Phi(\vec{X}_{i_1}) - \Phi(\vec{X}_{i_2}))Z_{i_2}}. \tag{4}$$

Another popular nonparametric density estimator is the projection density estimator. Given a series of orthonormal basis functions $\psi_l(\cdot)$, $l = 1, \ldots, \infty$, $f_T^\Phi(\vec{U})$ can be decomposed as $f_T^\Phi(\vec{U}) = \sum_{l=1}^\infty r_l \psi_l(\vec{U})$, where the coefficients are defined as $r_l = \int f_T^\Phi(\vec{U})\psi_l(\vec{U})d\vec{U}$. The projection method seeks to estimate $f_T^\Phi(\vec{U})$ with the first $L$ basis functions, i.e.,

$$\hat{f}_T^\Phi(\vec{U}) = \sum_{l=1}^L \hat{r}_l \psi_l(\vec{U}), \qquad \text{where } \hat{r}_l = \frac{1}{n_1} \sum_{i:Z_i=1} \psi_l(\vec{U}_i).$$

The projection density estimator then lead to the final estimator of $\mu_{CT}$

$$\hat{\mu}_{CT} = \frac{\sum_{i_1,i_2=1}^{n} Y_{i_1}(1 - Z_{i_1})K_L(\Phi(\vec{X}_{i_1}), \Phi(\vec{X}_{i_2}))Z_{i_2}}{\sum_{i_1,i_2=1}^{n}(1 - Z_{i_1})K_L(\Phi(\vec{X}_{i_1}), \Phi(\vec{X}_{i_2}))Z_{i_2}}, \tag{5}$$

where $K_L(\vec{x}, \vec{y}) = \sum_{l=1}^{L} \psi_l(\vec{x})\psi_l(\vec{y})$ denotes a projection kernel defined by the orthonormal basis $\{\psi_l(\cdot) : l = 1, \ldots, L\}$ (Giné & Nickl, 2016).

Although Algorithm 1 seems to suggest that $\hat{f}_T^\Phi$ is designed to estimate $f_T^\Phi$ at first glance, we would like to emphasize again that our ultimate goal is to estimate $\mu_{CT}$ instead of $f_T^\Phi$, so the choice of tuning parameters, $H$ in equation 4 and $L$ in equation 5, shall rely on the bias and variance trade-off in the final estimator $\hat{\mu}_{CT}$. The straightforward form of the estimators for $\mu_{CT}$ after applying the uniform transformer allows a more accurate trade-off between the bias and variance in estimating $\mu_{CT}$. We leave the detailed discussion of the tuning parameters $H$ and $L$ to Section 4.

### 3.2 ROSENBLATT'S UNIFORM TRANSFORMER WITH EMPIRICAL DENSITIES

There are various options to transform a distribution into a uniform distribution. In this section, we focus on a uniform transformer proposed by Rosenblatt (1952). More concretely, given the density of covariates in the control group, $f_C(\vec{X})$, we consider the following transformation $\Phi : \Omega \to [0,1]^d$

$$\begin{aligned}
\Phi(\vec{x})_{(1)} &= \mathbb{P}_C(X_{(1)} \le x_{(1)}), \\
\Phi(\vec{x})_{(2)} &= \mathbb{P}_C(X_{(2)} \le x_{(2)}|X_{(1)} = x_{(1)}), \\
&\vdots \\
\Phi(\vec{x})_{(d)} &= \mathbb{P}_C(X_{(d)} \le x_{(d)}|X_{(d-1)} = x_{(d-1)}, \ldots, X_{(1)} = x_{(1)}),
\end{aligned} \tag{6}$$

where $\vec{x} = (x_{(1)}, \ldots, x_{(d)}) \in \mathbb{R}^d$ is a vector, $\vec{X} = (X_{(1)}, \ldots, X_{(d)})$ is a random vector with density $f_C(\vec{X})$ and the corresponding probability $\mathbb{P}_C$. When $\vec{X}$ is a continuous random vector, $\Phi(\vec{X})$ follows a uniform distribution on $[0,1]^d$. It is worth noting that the uniform transformer relies on the condition that $\vec{X}$ is continuous. If some component of $\vec{X}$ is discrete, a random perturbation can be added to the observed covariates in the preprocessing step, as discussed in Brockwell (2007). In the following discussion, we assume $\vec{X}$ is a continuous random vector; for more discussions on discrete random variables, see Section C.2 in the Appendix. Clearly, the Rosenblatt's uniform transformer relies on the choice of the order $X_{(1)}, \ldots, X_{(d)}$. In practical applications, one could randomly select permutations of the orders and take the average of results.

In practice, we usually do not have much knowledge about the density $f_C(\vec{X})$, so we have no access to the uniform transformer $\Phi$ defined in equation 6 and need to construct the uniform transformer from the data. One natural way to construct the uniform transformer is to first estimate the density of covariates in the control group $f_C(\vec{X})$ by some estimator $\tilde{f}_C(\vec{X})$ and then define the uniform transformer based on $\tilde{f}_C(\vec{X})$ following the transformation in equation 6.

Furthermore, the construction of uniform transformers becomes easier if the covariates have a special correlation structure. For instance, equation 6 suggests that the uniform transformer $\Phi$ only relies on each marginal distribution of $f_C(\vec{X})$ if components of $\vec{X}$ are mutually independent. In other words, suppose $f_C(\vec{X})$ can be decomposed as $f_C(\vec{X}) = f_{C,1}(X_{(1)}) \times \ldots \times f_{C,d}(X_{(d)})$, then it is sufficient to construct the uniform transformer by estimating each marginal distribution. In this special case, we refer to the uniform transformer as a marginal uniform transformer. This idea can also be extended to densities with a group-wise mutually independent structure.

Moreover, the construction of the uniform transformer in WUNT only relies on the covariates of control samples. This feature enables the integration of a substantial amount of extra 'unlabeled' control samples (outcome of interest is not observed) which is available in a lot of applications, such as analysis of electronic health record data (Gronsbell & Cai, 2018; Chakrabortty & Cai, 2018). In such scenarios, a large amount of unlabeled data can be used to accurately estimate the density $f_C(\vec{X})$ using any suitable density estimator. Subsequently, the uniform transformer can be constructed based on this estimate and applied to the 'labeled' data set to compute the weights designed for treatment effects estimation.

### 3.3 Adaptive Uniform Transformer

Section 3.2 mainly focuses on Rosenblatt's uniform transformer and its empirical version with a density estimator. We provide a different angle to construct an empirical version of Rosenblatt's uniform transformer in this section. To illustrate the idea, we start with a one-dimensional case. When $d = 1$, Rosenblatt's transformation is defined by the cumulative distribution function of $\vec{X}$ in the control group. The cumulative distribution function can naturally be estimated by its empirical distribution, i.e., $\hat{\mathbb{P}}_{C,n}(\vec{x}) = \frac{1}{n_0} \sum_{i:Z_i=0} \vec{I}(\vec{X}_i \leq \vec{x})$, where $\vec{I}(\cdot)$ is an indicator function. If we plug $\hat{\mathbb{P}}_{C,n}$ in Rosenblatt's transformation $\Phi$, the resulting transformation maps $\{\vec{X}_i : Z_i = 0\}$ to $\{1/n_0, \ldots, 1\}$. In other words, $\hat{\mathbb{P}}_{C,n}$ help transform $\{\vec{X}_i : Z_i = 0\}$ to the grid points between 0 and 1. The benefit of this transformation is that it does not rely on estimating $f_C(\vec{X})$, so it is computationally simple. Can we construct a uniform transformer in a similar fashion for a multi-dimensional case?

To apply this idea for a $d$-dimensional covariate $\vec{X}$, we need to partition the data points evenly to $d$-dimensional grids. Without loss of generality, we assume the support of density is $\Omega = [0, 1]^d$ and $n_0 = N_0^d$ for some positive integer $N_0$ in this section. Based on $\{\vec{X}_i\}_{i:Z_i=0}$, we define the following data-driven partition of $\Omega$, $\Omega = \bigcup_{j_1,\ldots,j_d=1}^{N_0} Q_{j_1,j_2,\ldots,j_d} = \bigcup_{j_1,\ldots,j_d=1}^{N_0} I_{j_1} \times I_{j_1,j_2} \times \ldots \times I_{j_1,j_2,\ldots,j_d}$. Here, each $Q_{j_1,j_2,\ldots,j_d}$ is a cube and each $I_{j_1,j_2,\ldots,j_k}$ for $k \leq d$ is an interval. We construct the data-driven interval $I_{j_1,j_2,\ldots,j_k}$ in a hierarchical way. We first construct $\{I_{j_1}\}_{j_1=1}^{N_0}$, which is a partition of $[0, 1]$ such that there are exactly $N_0^{d-1}$ points in each $I_{j_1} \times [0, 1]^{d-1}$. After construction of $\{I_{j_1}\}_{j_1=1}^{N_0}$, we are ready to construct $\{I_{j_1,j_2}\}_{j_1,j_2=1}^{N_0}$. For each $j_1$, $\{I_{j_1,j_2}\}_{j_2=1}^{N_0}$ is a partition of $[0, 1]$ such that there are exactly $N_0^{d-2}$ points in each $I_{j_1} \times I_{j_1,j_2} \times [0, 1]^{d-2}$. The rest of $I_{j_1,j_2,\ldots,j_k}$ can be defined in a similar way. In doing so, each cube $Q_{j_1,j_2,\ldots,j_d}$ contains exactly one point. The idea is illustrated with an example of 9 data points on $[0, 1]^2$ in Figure 1 in Appendix A.

Then, we can easily approximate the cumulative distribution function and conditional cumulative distribution functions based on the partition to form an empirical Rosenblatt's uniform transformer. With this construction, we can put exactly mass $1/n_0$ for each cube $Q_{j_1,j_2,\ldots,j_d}$. The formal result is summarized in the following proposition. The proof is included in Appendix D.

**Proposition 1.** *Let $\hat{\Phi}_D$ be the uniform transformer defined in equation 6 by replacing $f_C(\vec{X})$ with $\ddot{f}(\vec{X}) = \frac{1}{|Q_{j_1,\ldots,j_d}| n_0} S\left(\frac{X_{(1)} - M(I_{j_1})}{|I_{j_1}|}\right) \times \ldots \times S\left(\frac{X_{(d)} - M(I_{j_1,\ldots,j_d})}{|I_{j_1,\ldots,j_d}|}\right)$, for any $\vec{X} \in Q_{j_1,j_2,\ldots,j_d}$, where $\vec{X} = (X_{(1)}, \ldots, X_{(d)})$, $|\cdot|$ represent the volume of a cube or the length of an interval and $M(\cdot)$ is the middle point of an interval. Here, $S(\cdot)$ is a smooth kernel function defined on [-0.5,0.5] such that $S(-0.5) = S(0.5) = 0$, $S(x) > 0$ if $x \in (-0/5, 0.5)$ and $\int_{-0.5}^{0.5} S(x) dx = 1$. Then $\hat{\Phi}_D$ satisfies the following properties: 1) For each cube $Q_{j_1,j_2,\ldots,j_d}$, $1 \leq j_1, \ldots, j_d \leq N_0$, we have $\hat{\Phi}_D(Q_{j_1,j_2,\ldots,j_d}) = \left[\frac{j_1-1}{N_0}, \frac{j_1}{N_0}\right) \times \ldots \times \left[\frac{j_d-1}{N_0}, \frac{j_d}{N_0}\right)$; 2) $\hat{\Phi}_D$ is a smooth map.*

This proposition suggests that $\hat{\Phi}_D$ is able to map the covariates in the control group to an approximately uniform distribution on $[0, 1]^d$, and we call it an adaptive uniform transformer. In particular, when $\Omega$ is on the real line ($d = 1$), $\hat{\Phi}_D$ can been seen as a smoothed version of the empirical cumulative distribution function of $\{\vec{X}_i\}_{i:Z_i=0}$. When there is no integer $N_0$ such that $n_0 = N_0^d$, we can choose $N_0$ as the largest integer such that $N_0^d < n_0$ and follow a similar procedure as above to distribute the data points evenly in the grids.

## 4 Theoretical Properties

We now turn to analyze the theoretical properties of our newly proposed framework. In this section, we consider uniform transformers introduced in Section 3 (either a uniform transformer constructed with a separate data set or an adaptive uniform transformer) and study the performance of both kernel and projection density estimators. In particular, our investigation focuses on Hölder class (van der Vaart & Wellner, 1996), $\mathcal{H}^\alpha(\Omega) = \left\{f : \Omega \to \mathbb{R} \big| \|f\|_{\alpha,\mathcal{H}} \leq M\right\}$, where the norm $\|f\|_{\alpha,\mathcal{H}}$ is

defined as $\|f\|_{\alpha,\mathcal{H}} = \max_{|k|\leq\lfloor\alpha\rfloor}\sup_{\vec{x}\in\Omega}|D^k f(\vec{x})| + \max_{|k|=\lfloor\alpha\rfloor}\sup_{\vec{x}_1\neq\vec{x}_2\in\Omega}\frac{|D^k f(\vec{x}_1)-D^k f(\vec{x}_2)|}{\|\vec{x}_1-\vec{x}_2\|^{\alpha-\lfloor\alpha\rfloor}}$.
Here, $k = (k_1,\ldots,k_d)$ with $|k| = k_1 + \ldots + k_d$ and the differential operator is defined as $D^k = \frac{\partial^{|k|}}{\partial x_{(1)}^{k_1}\ldots\partial x_{(d)}^{k_d}}$. We assume the basis functions $\{\psi_l : l = 1,\ldots,\infty\}$ in the projection density estimator form an orthonormal basis and satisfy

$$|r_l| \leq M_1 l^{-(\alpha/d+1/2)} \qquad \text{and} \qquad \sup_{\vec{x}}|\psi_l(\vec{x})| \leq M_2\sqrt{l}, \qquad \text{for any } l, \tag{7}$$

where $M_1$ and $M_2$ are some constants and $r_l$ is the coefficient of some given function $g \in \mathcal{H}^\alpha([0,1]^d)$, i.e., $g(\vec{x}) = \sum_{l=1}^\infty r_l\psi_l(\vec{x})$. For example, the wavelet basis satisfies this property (Giné & Nickl, 2016; Liang, 2019). When $Z_i = 0$, we write $Y_i = \mu_C(\vec{X}_i) + \epsilon_i$, where $\mathbb{E}(\epsilon_i|\vec{X}_i) = 0$. Through this section, we assume

$$\mathbb{E}(\epsilon_i^2) := \sigma(\vec{X}_i)^2 \leq \sigma^2, \qquad \text{for some constant } \sigma^2. \tag{8}$$

We first investigate the performance of the proposed estimator when the uniform transformer in equation 6 is defined by some fixed density $\tilde{f}_C(\vec{X})$, which might be different from $f_C(\vec{X})$. The following theorem characterizes the convergence rate of the estimator.

**Theorem 1.** *Let $\hat{\mu}_{CT}$ be the estimator defined in equation 4 with an $\alpha + \beta$ order kernel or the one defined in equation 5 with basis function satisfying equation 7. Suppose the uniform transformer is defined in equation 6 with some density $\tilde{f}_C(\vec{X})$. Assume $\mu_C^\Phi \in \mathcal{H}^\alpha([0,1]^d)$, $f_T^\Phi \in \mathcal{H}^\beta([0,1]^d)$ and $f_C^\Phi \in \mathcal{H}^\gamma([0,1]^d)$ with $0 < \alpha,\beta < \gamma$. We further assume conditions equation 1 and equation 8 hold and $f_T(\vec{X}) = 0$ when $\vec{X}$ is at boundary of $[0,1]^d$ if we use estimator defined in equation 4. If we choose $h_1 = \ldots = h_d = h = n^{-2/(d+2(\alpha+\beta))}$ in equation 4 or $L = n^{2d/(d+2(\alpha+\beta))}$ in equation 5, then there exists a constant $C_0$ such that*

$$\mathbb{E}(\hat{\mu}_{CT} - \mu_{CT})^2 \leq C_0\left(n^{-\frac{4(\alpha+\beta)}{d+2(\alpha+\beta)}} + n^{-1} + \Delta^2\right),$$

*where $\Delta$ is the difference between $\tilde{f}_C(\vec{X})$ and $f_C(\vec{X})$ in $L^2$ norm, i.e., $\Delta = \|\tilde{f}_C(\vec{X}) - f_C(\vec{X})\|_2$.*

Theorem 1 suggests that the performance of new estimators depends on the sum of smoothness levels of the response and density functions. Notably, they can still work well even when the response function is non-smooth ($\alpha < d/2$). In addition, it is worth noting that the optimal choice of tuning parameter $h$ or $L$ relies on the levels of smoothness of both the response and density functions. The optimal choice for estimating $f_T^\Phi$ ($h = n^{-1/(d+2\beta)}$ or $L = n^{d/(d+2\beta)}$) can lead to a suboptimal convergence rate in estimating $\mu_{CT}$. In other words, the best way to estimate $f_T^\Phi$ (hence the weights $w_i$) may not necessarily lead to the best estimator for $\mu_{CT}$.

An immediate result of Theorem 1 characterizes the performance of the proposed estimators when the density of covariates in the control group $f_C(\vec{X})$ is known. Specifically, with a known $f_C(\vec{X})$, the convergence rate of the estimator in equation 4 or equation 5 is

$$n^{-\frac{4(\alpha+\beta)}{d+2(\alpha+\beta)}} + n^{-1}. \tag{9}$$

If the density of covariates in the control group $f_C(\vec{X})$ is not known, it can be estimated by some density estimator $\tilde{f}_C(\vec{X})$ with a separate data set, as discussed in Section 3.2. In this case, the uniform transformer is constructed with this separate data set. The following corollary can further characterize the performance of the new estimators.

**Corollary 1.** *Let $\tilde{f}_C(\vec{X})$ in Theorem 1 be a density estimated by $N = cn^t$ control samples for some $t \geq 1$ and constant $c > 0$. If $\|\tilde{f}_C(\vec{X}) - f_C(\vec{X})\|_2 \leq N^{-\kappa/(d+2\kappa)}$ for some constant $\kappa$, we have the following results. When $\alpha + \beta \leq d/2$ and $\kappa > 2(\alpha+\beta)d/(td + (2t-4)(\alpha+\beta))$, then*

$$\mathbb{E}(\hat{\mu}_{CT} - \mu_{CT})^2 \leq Cn^{-\frac{4(\alpha+\beta)}{d+2(\alpha+\beta)}}.$$

*When $\alpha + \beta > d/2$ and $\kappa > d/2(t-1)$, then*

$$\frac{\sqrt{n}(\hat{\mu}_{CT} - \mu_{CT})}{\sqrt{V}} \to N(0,1),$$

*where $N(0,1)$ is standard normal distribution, $P = \mathbb{P}(Z = 1)$ and the variance $V$ is $V = \int\frac{\mu_C^2(\vec{x})f_T(\vec{x})}{P}d\vec{x} + \int\frac{(\sigma^2(\vec{x})+\mu_C^2(\vec{x}))f_T^2(\vec{x})}{(1-P)f_C(\vec{x})}d\vec{x} - 4\left(\int\mu_C(\vec{x})f_T(\vec{x})d\vec{x}\right)^2$.*

We can conclude from this corollary that the converge rate in equation 9 is still achievable as long as $f_C(\vec{X})$ can be estimated accurately. Similar to many other popular weighting methods (Hahn, 1998; Hirano et al., 2003; Chan et al., 2016; Fan et al., 2023), $\hat{\mu}_{CT}$ is a $\sqrt{n}$-consistent estimator when $\alpha + \beta > d/2$. Now, we show that the converge rate in equation 9 is actually sharp in terms of minimax optimality. More specifically, we consider the following family of data distribution $(\vec{X}, Z, Y) \sim F$ in $\mathcal{F}_{\alpha,\beta} := \{F : \mu_C^\Phi \in \mathcal{H}^\alpha([0,1]^d), f_T^\Phi \in \mathcal{H}^\beta([0,1]^d) \text{ and } equation\ 1, equation\ 8 \text{ hold}\}$.

**Theorem 2.** *Assume $f_C(\vec{X})$ is known in advance so that $\Phi$ is defined based on $f_C(\vec{X})$. Consider estimating $\mu_{CT}$ on $\mathcal{F}_{\alpha,\beta}$ with $\alpha, \beta > 0$. Then there exists a constant $c_0$ such that*

$$\inf_{\hat{\mu}_{CT}} \sup_{F \in \mathcal{F}_{\alpha,\beta}} \mathbb{E}(\hat{\mu}_{CT} - \mu_{CT})^2 \geq c_0 \left( n^{-\frac{4(\alpha+\beta)}{d+2(\alpha+\beta)}} + n^{-1} \right).$$

Theorem 2 characterizes the difficulty of estimating the treatment effects in the ideal case when $f_C(\vec{X})$ is known. In general, if $f_C(\vec{X})$ is unknown, the problem becomes harder, so the convergence rate will be lower bounded by the result in Theorem 2. Notably, Theorems 1 and 2 together show that the converge rate in equation 9 is the minimax optimal for estimating $\mu_{CT}$ with a known $f_C(\vec{X})$. That is, if we know the density $f_C(\vec{X})$ beforehand, the minimax optimal rate of estimating $\tau_{\text{ATT}}$ is

$$\inf_{\hat{\tau}_{\text{ATT}}} \sup_{F \in \mathcal{F}_{\alpha,\beta}} \mathbb{E}(\hat{\tau}_{\text{ATT}} - \tau_{\text{ATT}})^2 \asymp n^{-\frac{4(\alpha+\beta)}{d+2(\alpha+\beta)}} + n^{-1}.$$

We now show that the new estimator is still reliable without assumptions on $f_C(\vec{X})$ if the uniform transformer is constructed as in Proposition 1.

**Theorem 3.** *Let $\hat{\mu}_{CT}$ be the estimator defined in equation 4 with an $\alpha + \beta$ order kernel or the one defined in equation 5 with basis function satisfying equation 7. Assume the uniform transformer is defined in Proposition 1. Suppose $\mu_C^\Phi \in \mathcal{H}^\alpha([0,1]^d)$, $f_T^\Phi \in \mathcal{H}^\beta([0,1]^d)$ with arbitrary $\alpha, \beta > 0$ and conditions equation 1, equation 8 hold. For the kernel estimator, we choose bandwidth $h_1 = \ldots = h_d = h$ satisfying $n^2 h^d \to \infty$ and $h \to 0$. For the projection estimator, we choose the number of basis $L$ satisfying $n^2 L^{-1} \to \infty$ and $L \to \infty$. Then,*

$$\hat{\mu}_{CT} \to_p \mu_{CT}, \qquad \text{as } n \to \infty.$$

Theorem 3 shows that this new uniform transformer can help build a consistent treatment effect estimator with no assumption on $f_C(\vec{X})$ and very mild conditions on $f_T^\Phi$ and $\mu_C^\Phi$. Proofs can be found in Appendix D. In this section, we focus mainly on the estimation of $\mu_{CT}$. All these results can naturally lead to the conclusion for the average treatment effect on the treated group, $\tau_{\text{ATT}}$. Besides the theoretical results, we also conduct the numerical experiments in Section B of the Appendix.

## 5 CONCLUDING REMARKS

In this paper, we propose a novel framework of weighting methods for treatment effects estimation, named Weighting by Uniform Transformer (WUNT). Unlike the existing weighting methods, WUNT employs a data-driven uniform transformer to the observed covariates, effectively transforming the control group's covariate distribution to a uniform distribution. This transformation allows us to design the weights in WUNT specifically tailored for treatment effects estimation, resulting in a straightforward kernel-based $U$-statistic as the final estimator. We delve into the theoretical properties of the newly proposed framework under a nonparametric setting. Our investigation shows that, with weights chosen by WUNT, the weighting method can achieve the minimax optimal rate of estimating the average treatment effect on the treated group, even under low regularity conditions. Notably, the tuning parameter in WUNT needs to be chosen based on the smoothness levels of both the response and density functions to achieve an accurate trade-off between bias and variance under low regularity conditions. In cases where the smoothness levels are unknown, Lepski method (Lepski, 1991; 1992) can help WUNT select the tuning parameter in a data-driven way. Additional implementation guidelines are provided in Section C.3 of the Appendix.

ACKNOWLEDGMENTS

The authors are grateful for the helpful comments from Paul R. Rosenbaum, Dylan S. Small, Eric J. Tchetgen Tchetgen and Jose R. Zubizarreta. S. Wang would like to acknowledge support for this project from the National Science Foundation (NSF grant DMS-2113458).

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

# A   APPENDIX – EXAMPLE FOR THE CONSTRUCTION OF ADAPTIVE UNIFORM TRANSFORMER

In this section, we illustrate the construction of an adaptive uniform transform using a toy example with 9 data points on $[0, 1]^2$ in Figure 1.

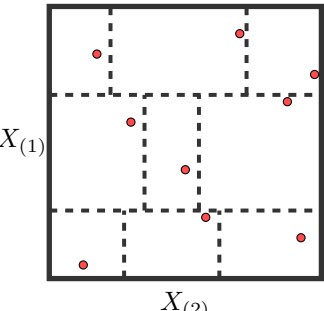

Figure 1: An illustrative example for the construction of an adaptive uniform transformer when $d = 2$.

# B   APPENDIX – NUMERICAL EXPERIMENTS

In this section, we study the numerical performance of our proposed framework by carrying out several simulation studies to estimate the average treatment effect on the treated group (ATT).

## B.1   COMPARISON OF UNIFORM TRANSFORMERS

In the first set of simulation studies, we compare four ways to construct the uniform transformer $\Phi$ from the control samples. More specifically, the adaptive uniform transformers are constructed in four different ways according to the following: if extra 'unlabeled' control samples are used

or not, and if the uniform transformers are based on joint or marginal distribution. To simulate the observed data, we draw $(W_1, ..., W_5) \sim N((0.5, ..., 0.5), \Sigma)$ in the treated group and $(W_1, ..., W_5) \sim N((0, ..., 0), \Sigma)$ in the control group, where $N$ represents the normal distribution and each entry of the covariance matrix $\Sigma$ is defined as $\Sigma_{ij} = \rho^{|i-j|}$. We vary $\rho$ from 0, 0.1, 0.2 and 0.3. The observed covariates of each sample is $\vec{X} = (X_1, ..., X_5)$ that $X_i = \exp(W_i) + W_i$. We consider two models for the outcome of interest: $Y_1 = W_1^2 W_2^2 - 2W_3^2 W_4^2 + \sum_{i=1}^5 W_i + \epsilon_1$ and $Y_2 = 10 \sum_{i=1}^3 W_i + 100 \prod_{i=1}^2 \sin(2\pi W_i) + 100 \prod_{i=3}^5 \cos(\pi W_i/2) + \epsilon_2$, where $\epsilon_i \sim N(0, 1)$ follows independent standard normal distribution. We consider two density estimators – the kernel density estimator and the projection density estimator, and denote their corresponding estimators of the ATT by $\hat{\tau}_{\text{ATT}}^K$ and $\hat{\tau}_{\text{ATT}}^P$, respectively. The sample size is 500 for the treated group and 1000 for the control group. The extra 'unlabeled' control samples are drawn in the same way with sample size 10000. The performances of different uniform transformers are evaluated by bias and root mean squared error (RMSE), calculated from 500 replications of simulation experiments.

The results are summarized in Table 1. These results show that when the covariate distribution of control samples is independent, the marginal uniform transformer works slightly better than the joint uniform transformer. This observation makes sense because when $f_C$ is an independent distribution, it is sufficient to construct the uniform transformer for each marginal distribution, as discussed in Section 3.2. On the other hand, the joint uniform transformer is more robust when the observed covariates are correlated. In addition, the uniform transformer works in a better way when extra data is available.

Table 1: Comparison of uniform transformers under different covariance matrices.

| | | | No extra data | | | | With extra data | | | |
| | | | Joint | | Marginal | | Joint | | Marginal | |
| | | $\rho$ | Bias | RMSE | Bias | RMSE | Bias | RMSE | Bias | RMSE |
|---|---|---|---|---|---|---|---|---|---|---|
| $Y_1$ | $\hat{\tau}_{\text{ATT}}^K$ | 0 | -0.09 | 0.49 | -0.11 | 0.47 | 0.06 | 0.48 | -0.03 | 0.45 |
| | | 0.1 | -0.16 | 0.56 | -0.26 | 0.59 | 0.00 | 0.53 | -0.18 | 0.54 |
| | | 0.2 | -0.22 | 0.64 | -0.36 | 0.74 | -0.06 | 0.60 | -0.3 | 0.67 |
| | | 0.3 | -0.30 | 0.71 | -0.36 | 0.88 | -0.13 | 0.68 | -0.37 | 0.80 |
| | $\hat{\tau}_{\text{ATT}}^P$ | 0 | 0.18 | 0.56 | -0.06 | 0.51 | 0.03 | 0.53 | -0.06 | 0.52 |
| | | 0.1 | 0.13 | 0.61 | -0.11 | 0.58 | 0.00 | 0.58 | -0.11 | 0.58 |
| | | 0.2 | 0.11 | 0.68 | -0.14 | 0.66 | -0.01 | 0.66 | -0.14 | 0.66 |
| | | 0.3 | 0.09 | 0.75 | -0.16 | 0.75 | -0.02 | 0.74 | -0.16 | 0.75 |
| $Y_2$ | $\hat{\tau}_{\text{ATT}}^K$ | 0 | 1.15 | 4.31 | 1.46 | 4.16 | 1.74 | 4.36 | 0.82 | 3.96 |
| | | 0.1 | 1.24 | 4.22 | 0.47 | 4.08 | 1.84 | 4.43 | 0.28 | 3.98 |
| | | 0.2 | 1.13 | 4.14 | -0.82 | 4.17 | 1.77 | 4.32 | -0.55 | 4.02 |
| | | 0.3 | 0.88 | 4.05 | -2.31 | 4.61 | 1.46 | 4.10 | -1.59 | 4.19 |
| | $\hat{\tau}_{\text{ATT}}^P$ | 0 | 2.19 | 4.61 | 1.98 | 4.70 | 1.83 | 4.48 | 1.96 | 4.69 |
| | | 0.1 | 2.24 | 4.57 | 1.86 | 4.57 | 1.74 | 4.42 | 1.84 | 4.56 |
| | | 0.2 | 2.23 | 4.57 | 1.71 | 4.44 | 1.65 | 4.31 | 1.69 | 4.43 |
| | | 0.3 | 2.13 | 4.43 | 1.51 | 4.19 | 1.55 | 4.16 | 1.49 | 4.18 |

## B.2 COMPARISON OF ATT ESTIMATORS

The second set of simulation studies compares the newly proposed estimators with other existing methods under the above model. The four new estimators we consider here are: uniform transformer on joint distribution + kernel density estimator, uniform transformer on marginal distribution + kernel density estimator, uniform transformer on joint distribution + projection density estimator, and uniform transformer on marginal distribution + projection density estimator. We compare them with the inverse probability weighting estimator (IPW) with the propensity score estimated by random forests with R package `randomForest`, covariate balancing propensity score (CBPS) proposed by Imai & Ratkovic (2014) with R package `CBPS`, empirical balancing calibration weighting (CAL) by Chan et al. (2016) with R package `ATE`, and stable weights (SBW) proposed by Zubizarreta (2015) with R package `sbw`. We still adopt bias and RMSE, calculated from 500 replications of simulation experiments again, as our measure of performance of these estimators for ATT. The data is generated in the same way as the first set of simulation studies with $\rho = 0$. Results summarized in Table 2

suggest that our new proposed estimators perform better than the other methods in terms of bias and RMSE.

Table 2: Comparison of different ATT estimators on model $Y_1$ and $Y_2$.

| | $Y_1$ | | $Y_2$ | |
| --- | --- | --- | --- | --- |
| | Bias | RMSE | Bias | RMSE |
| Kernel+Joint | -0.10 | 0.51 | 0.94 | 4.38 |
| Kernel+Marginal | -0.11 | 0.47 | 1.29 | 4.21 |
| Projection+Joint | 0.20 | 0.58 | 2.01 | 4.57 |
| Projection+Marginal | -0.04 | 0.53 | 1.85 | 4.65 |
| IPW | 0.71 | 0.85 | 6.18 | 7.15 |
| CBPS | 1.11 | 1.96 | 3.49 | 6.00 |
| CAL | 1.07 | 1.64 | 3.93 | 5.99 |
| SBW | 0.45 | 0.84 | 3.4 | 5.41 |

### B.3 COMPARISON OF ATT ESTIMATORS WITH DIFFERENT SAMPLE SIZES

In the third set of simulation studies, we further compare the eight estimators of ATT in the second set of simulation studies and assess their performance with different sample sizes. In this set of simulation experiments, the data is simulated based on the example in Kang & Schafer (2007). More concretely, we draw $W = (W_1, W_2, W_3, W_4)$ from $N((0, 0, 0, 0), I)$, where $I$ is a $4 \times 4$ identity matrix and consider the following two models of the outcome of interest: $Y_3 = 210 + 27.4W_1 + 13.7W_2 + 13.7W_3 + 13.7W_4 + \epsilon_3$ (the same with Kang & Schafer (2007)) and $Y_4 = (4W_1 + 2W_2)/(\exp(W_3) + 4\sqrt{|W_4|}) + 2W_3 + W_4 + \epsilon_4$. Here, $\epsilon_i$ also follows independent standard normal distribution. Each sample is assigned to the treated group with probability (i.e., true propensity score) $1/(1 + \exp(W_1 - 0.5W_2 + 0.252_3 + 0.1W_4))$. Instead of observing the covariates $W$, we are able to observe only the transformed data $X_1 = \exp(W_1/2)$, $X_2 = W_2/(1 + \exp(W_1)) + 10$, $X_3 = (W_1 W_3/25 + 0.6)^3$ and $X_4 = (W_2 + W_4 + 20)^2$. In order to assess the effect of sample size, we vary it from 1000, 2000, and 5000. Similar to the previous two simulation studies, bias and RMSE based on 500 replications of simulation experiments are summarized in Table 3. The results show that our new estimators generally outperform other existing methods. The only exception is the kernel density estimator equipped with a uniform transformer based on marginal distribution. The reason is that the observed covariates are highly dependent and the kernel density estimator seems to be sensitive to the correlation among covariates. Table 3 also suggests that the new estimators based on the kernel density estimator can constantly reduce the bias as the sample size increases.

### B.4 COMPARISON OF COMPUTATION COMPLEXITY

We compare the computation time of these eight methods in the last set of simulation studies. In particular, we consider two optimization solvers for SBW: quadpros (the default choice) and mosek (a commercial solver available from `https://www.mosek.com/`). We record the average time from 10 replications of the model $Y_3$ with sample sizes 1000, 2000, 5000, and 10000 and summarize them in Table 4. All these algorithms are evaluated with the same laptop (Intel Core i5 @2.3 GHz/8GB). From Table 4, we can conclude that the new estimators based on the projection density estimator can be computed efficiently. The main computation obstacle of the new estimators based on the kernel density estimator is the kernel $U$ statistics which have $O(n^2)$ computation complexity. It is also interesting to note that the uniform transformer can be constructed in a very short time ($< 0.66$s even when $n = 10000$) and thus can be applied on a larger scale dataset.

## C APPENDIX – DISCUSSIONS

### C.1 AVERAGE TREATMENT EFFECT

In the paper, we mainly focus on the estimation of average treatment effect on the treated group. All the methodology and theoretical properties can be readily generalized to average treatment effect.

Table 3: Comparison of different ATT estimators on model $Y_3$ and $Y_4$.

| $Y_3$ | $n = 1000$ | | $n = 2000$ | | $n = 5000$ | |
|---|---|---|---|---|---|---|
| | Bias | RMSE | Bias | RMSE | Bias | RMSE |
| Kernel+Joint | -7.90 | 8.21 | -6.32 | 6.51 | -4.50 | 4.62 |
| Kernel+Marginal | -9.88 | 10.16 | -9.46 | 9.62 | -8.88 | 8.96 |
| Projection+Joint | -4.26 | 4.48 | -4.39 | 4.49 | -4.27 | 4.30 |
| Projection+Marginal | -4.00 | 4.15 | -4.01 | 4.09 | -3.98 | 4.00 |
| IPW | -10.13 | 10.26 | -9.96 | 10.02 | -9.61 | 9.64 |
| CBPS | -5.35 | 5.56 | -5.40 | 5.51 | -5.31 | 5.37 |
| CAL | -4.36 | 4.49 | -4.43 | 4.49 | -4.37 | 4.40 |
| SBW | -7.22 | 7.32 | -7.40 | 7.44 | -7.4 | 7.42 |

| $Y_4$ | $n = 1000$ | | $n = 2000$ | | $n = 5000$ | |
|---|---|---|---|---|---|---|
| | Bias | RMSE | Bias | RMSE | Bias | RMSE |
| Kernel+Joint | -0.40 | 0.44 | -0.31 | 0.34 | -0.22 | 0.24 |
| Kernel+Marginal | -0.67 | 0.71 | -0.66 | 0.68 | -0.65 | 0.66 |
| Projection+Joint | -0.38 | 0.41 | -0.38 | 0.39 | -0.37 | 0.38 |
| Projection+Marginal | -0.38 | 0.4 | -0.38 | 0.39 | -0.39 | 0.39 |
| IPW | -0.63 | 0.64 | -0.61 | 0.62 | -0.58 | 0.59 |
| CBPS | -0.56 | 0.59 | -0.57 | 0.59 | -0.57 | 0.58 |
| CAL | -0.46 | 0.49 | -0.47 | 0.49 | -0.48 | 0.48 |
| SBW | -0.64 | 0.66 | -0.66 | 0.67 | -0.66 | 0.67 |

Table 4: Comparison of different ATT estimators in terms of the computation time, which is shown in seconds.

| | $n = 1000$ | $n = 2000$ | $n = 5000$ | $n = 10000$ |
|---|---|---|---|---|
| Kernel+Joint | 0.40 | 1.36 | 7.98 | 31.08 |
| Kernel+Marginal | 0.36 | 1.31 | 8.01 | 32.19 |
| Projection+Joint | 0.06 | 0.12 | 0.30 | 0.66 |
| Projection+Marginal | 0.04 | 0.11 | 0.49 | 1.85 |
| IPW | 0.33 | 0.73 | 2.24 | 4.61 |
| CBPS | 0.31 | 0.64 | 1.67 | 4.23 |
| CAL | 0.14 | 0.27 | 0.67 | 1.41 |
| SBW(quadpros) | 0.14 | 1.19 | 20.60 | 152.18 |
| SBW(mosek) | 0.03 | 0.04 | 0.08 | 0.19 |

Recall the average treatment effect is

$$\tau_{\text{ATE}} = \mu_{TA} - \mu_{CA} = \int \mu_T(\vec{X})f(\vec{X})d\vec{X} - \int \mu_C(\vec{X})f(\vec{X})d\vec{X}.$$

In order to estimate $\mu_{TA}$ and $\mu_{CA}$ separately, we can construct two uniform transformers based on treated samples $\{\vec{X}_i\}_{i:Z_i=1}$ and control samples $\{\vec{X}_i\}_{i:Z_i=0}$, respectively, which are named $\Phi_T$ and $\Phi_C$. After applying transformation $\Phi_T$ and $\Phi_C$ on all data, we can pick suitable density estimators for $f^{\Phi_T}(\vec{X})$ and $f^{\Phi_C}(\vec{X})$, which are density of $\Phi_T(\vec{X})$ and $\Phi_C(\vec{X})$. The corresponding algorithm is summarized in Algorithm 2. For example, if we adopt the projection density estimator, the resulting estimator is

$$\hat{\tau}_{\text{ATE}} = \frac{\sum_{i_1 \neq i_2} Y_{i_1} Z_{i_1} K_L(\Phi_T(\vec{X}_{i_1}), \Phi_T(\vec{X}_{i_2}))}{\sum_{i_1 \neq i_2} Z_{i_1} K_L(\Phi_T(\vec{X}_{i_1}), \Phi_T(\vec{X}_{i_2}))} - \frac{\sum_{i_1 \neq i_2} Y_{i_1} (1 - Z_{i_1}) K_L(\Phi_C(\vec{X}_{i_1}), \Phi_C(\vec{X}_{i_2}))}{\sum_{i_1 \neq i_2} (1 - Z_{i_1}) K_L(\Phi_C(\vec{X}_{i_1}), \Phi_C(\vec{X}_{i_2}))}.$$

## C.2 UNIFORM TRANSFORMER FOR DISCRETE VARIABLES

Most of this paper focuses on when the observed covariates are continuous random variables. However, we also observe some discrete or categorical covariates in practice. Motivated by Brockwell (2007), we can first transform the discrete random variable to a continuous random variable. For the

---

**Data:** $\{(\vec{X}_i, Z_i, Y_i)\}_{i=1}^n$.
**Result:** Weights $w_i$ and an estimator of $\tau_{\text{ATE}}$.
Construct the uniform transformer $\Phi_T$ and $\Phi_C$ by $\{\vec{X}_i\}_{i:Z_i=1}$ and $\{\vec{X}_i\}_{i:Z_i=0}$;
Estimate $f^{\Phi_T}(\vec{X})$ and $f^{\Phi_C}(\vec{X})$ from $\{\Phi_T(\vec{X}_i)\}$ and $\{\Phi_C(\vec{X}_i)\}$;
Evaluate the weights by

$$
w_i = \begin{cases} -\hat{f}^{\Phi_C}(\Phi_C(\vec{X}_i))/\sum_{i:Z_i=0} \hat{f}^{\Phi_C}(\Phi_C(\vec{X}_i)) & Z_i = 0 \\ \hat{f}^{\Phi_T}(\Phi_T(\vec{X}_i))/\sum_{i:Z_i=1} \hat{f}^{\Phi_T}(\Phi_T(\vec{X}_i)) & Z_i = 1 \end{cases}
$$

Estimate $\tau_{\text{ATE}}$ by

$$
\hat{\tau}_{\text{ATE}} = \sum_{i=1}^n w_i Y_i
$$

**Algorithm 2**. Weighting by Uniform Transformer for Average Treatment Effect

---

sake of concreteness, we assume $\vec{X}$ is a binary variable in this section, i.e., $\vec{X} \in \{0,1\}$. In order to construct a continuous variant of $\vec{X}$, we add a random perturbation to $\vec{X}$

$$
\tilde{\vec{X}} = \begin{cases} \vec{X} + R & \vec{X} = 0 \\ \vec{X} + aR & \vec{X} = 1 \end{cases},
$$

where $R$ is an independent uniform random variable on $[0,1]$ and $a$ is some positive number. It is clear that $\tilde{\vec{X}}$ is a continuous random variable and satisfies strong ignorability of the treatment assignment in equation 1 if $\vec{X}$ satisfies it. If we choose $a = \mathbb{P}(\vec{X} = 1)/\mathbb{P}(\vec{X} = 0)$, the resulting distribution of $\tilde{\vec{X}}$ is also a smooth one. Therefore, it is sufficient to consider a uniform transformer constructed by $\tilde{\vec{X}}$. It is clear that the resulting uniform transformer from data $\tilde{\vec{X}}$ and the resulting weight and estimator highly rely on particular realization of $R$. In order to reduce the effect of particular realization of $R$, we also suggest considering multiple independent copies of $R$ (i.e., independent copies of $\tilde{\vec{X}}$ given $\vec{X}$), evaluating the weight on each copy of $\tilde{\vec{X}}$ and taking the average of these weights as the final weight.

Based on the above discussion, we consider a very simple preprocessing step for discrete variable in practice. Specifically, we transform the discrete variable to its rank by breaking ties randomly. In this way, we obtain a similar variant of $\tilde{\vec{X}}$ with $a = \#\{\vec{X} = 1\}/\#\{\vec{X} = 0\}$, where $\#$ represents the number of elements in a set. Then, we can regard these ranks as continuous random variable and evaluate uniform transformer and weights on them directly. Again, in order to reduce the effect of randomly breaking ties, we can repeat the process multiple times and take the average of the results.

### C.3 IMPLEMENTATION SUGGESTIONS

We present several uniform transformers and density estimators in Section 3 and now would like to give some practical suggestions on them. In terms of uniform transformer, we recommend the newly proposed adaptive uniform transformer defined in Proposition 1, because a) its computational complexity is $O(n)$; b) there is no need for sample splitting and c)the resulting estimator does not rely on strong assumptions of $f_C(\vec{X})$. We can apply the adaptive uniform transformer in two different ways based on the knowledge of $f_C(\vec{X})$. More concretely, if we know $f_C(\vec{X})$ is approximately mutually independent, then construction of the adaptive uniform transformer based on marginal distributions is suggested. On the other hand, if the distribution is not mutually independent, we suggest constructing the adaptive uniform transformer based on the joint distribution directly.

We can choose different density estimators according to our knowledge of $\mu_C^{\Phi}(\vec{X})$ and $f_T^{\Phi}(\vec{X})$. If the specific form of response function and density is not known, we suggest the kernel density estimator. The reason is that a) its implementation is easy; b) there is no need to choose basis functions and c) it performs relatively robustly. If we know $\mu_C^{\Phi}(\vec{X})$ or $f_T^{\Phi}(\vec{X})$ can be represented by a few simple

basis functions in advance, we suggest choosing the projection estimator for density. Based on a few basis functions, the resulting projection estimator is simple and accurate, and can be rapidly computed. We summarize the choice of uniform transformers and density estimators in Table 5.

| | | Independence of $f_C$ | |
| --- | --- | --- | --- |
| | | Yes | No |
| Knowledge of $\mu_C^\Phi, f_T^\Phi$ | Yes | Marginal+Projection | Joint + Projection |
| | No | Marginal + Kernel | Joint + Kernel |

Table 5: Summary of implementation suggestion.

When smoothness levels are unknown, we introduce a data-driven way to select tuning parameter by following Lepski method (Lepski, 1991; 1992). See also Giné & Nickl (2008). Write $\hat{\mu}_{CT}(h)$ as the estimator defined in equation 4 when the bandwidth is $h$, $\sigma(h, n) = 1/(nh^{d/2})$ and $d(h) = \sqrt{\log(h_1/h)}$. Define a grid of bandwidths

$$\tilde{\mathcal{H}} = \left\{ h \in \left[ \frac{\log n}{n^{2/d}}, \frac{\log n}{n^{1/d}} \right] : h_1 = \frac{\log n}{n^{1/d}}, h_{k+1} = h_k/\rho, k = 1, 2, \ldots \right\},$$

where $\rho > 1$. The bandwidth estimator is

$$\hat{h} = \max \left\{ h \in \tilde{\mathcal{H}} : |\hat{\mu}_{CT}(h) - \hat{\mu}_{CT}(g)| \leq \sigma(g, n)d(g), \quad \forall g < h, g \in \tilde{\mathcal{H}} \right\}.$$

Similarly, we can also define a data-driven way to select the number of basis function in the estimator defined in equation 5, but omit the details.

## D   APPENDIX – PROOFS

To distinguish from the constants that appeared in the previous sections, we shall use the capital letter $C$ and lower case letter $c$ to denote generic positive constants that may take different values at each appearance.

### D.1   PROOF OF PROPOSITION 1

To study the properties of $\hat{\Phi}_D$, we first calculate the marginal distribution of $\ddot{f}(\vec{X})$. For any $X \in I_{j_1}$, the marginal distribution is

$$\ddot{f}^1(X) = \int \ddot{f}(\vec{X})dX_{(2)} \ldots dX_{(d)} = \frac{N_0^{d-1}}{|I_{j_1}|n_0} S \left( \frac{X - M(I_{j_1})}{|I_{j_1}|} \right).$$

Similarly, for any $\vec{X}^k = (X_{(1)}, \ldots, X_{(k)}) \in Q_{j_1, \ldots, j_k} := I_{j_1} \times I_{j_1, j_2} \times \ldots \times I_{j_1, \ldots, j_k}$, where $k \leq d$, the marginal distribution can be written as

$$\ddot{f}^k(\vec{X}^k) = \int \ddot{f}(\vec{X})dX_{(k+1)} \ldots dX_{(d)}$$

$$= \frac{N_0^{d-k}}{|Q_{j_1, \ldots, j_k}|n_0} \prod_{s=1}^{k} S \left( \frac{X_{(s)} - M(I_{j_1, \ldots, j_s})}{|I_{j_1, \ldots, j_s}|} \right).$$

It is clear that

$$\int_{Q_{j_1, \ldots, j_k}} \ddot{f}^k(\vec{X}^k)d\vec{X}^k = \frac{1}{N_0^k}.$$

Then, we can define its conditional cumulative distributional function

$$\ddot{F}_1(X_{(1)}) = \int_0^{X_{(1)}} \ddot{f}^1(X)dX$$

and

$$\ddot{F}_k(X_{(k)}|X_{(1)}, \ldots, X_{(k-1)}) = \int_0^{X_{(k)}} \frac{\ddot{f}^k(X_{(1)}, \ldots, X_{(k-1)}, X)}{\ddot{f}^{k-1}(X_{(1)}, \ldots, X_{(k-1)})}dX.$$

The transformation $\hat{\Phi}_D$ is then defined as

$$\hat{\Phi}_D(\vec{X}) = (\ddot{F}_1(X_{(1)}), \ldots, \ddot{F}_d(X_{(d)}|X_{(1)}, \ldots, X_{(d-1)})).$$

We are now ready to verify the first property. Suppose $\vec{X} \in Q_{j_1,\ldots,j_d}$. Since $X_{(1)} \in I_{j_1}$,

$$\frac{j_1 - 1}{N_0} = \sum_{r=1}^{j_1-1} \int_{I_r} \ddot{f}^1(X)dX \leq \ddot{F}_1(X_{(1)}) < \sum_{r=1}^{j_1} \int_{I_r} \ddot{f}^1(X)dX = \frac{j_1}{N_0}.$$

Similarly, because $X_{(k)} \in I_{j_1,\ldots,j_k}$,

$$\int_0^{X_{(k)}} \ddot{f}^k(X_{(1)}, \ldots, X_{(k-1)}, X)dX$$

$$< \sum_{r=1}^{j_k} \int_{I_{j_1,\ldots,j_{k-1},r}} \ddot{f}^k(X_{(1)}, \ldots, X_{(k-1)}, X)dX$$

$$= \sum_{r=1}^{j_k} \left( \frac{N_0^{d-k}}{|Q_{j_1,\ldots,j_{k-1}}|n_0} \prod_{s=1}^{k-1} S\left( \frac{X_{(s)} - M(I_{j_1,\ldots,j_s})}{|I_{j_1,\ldots,j_s}|} \right) \right)$$

$$= \frac{N_0^{d-k}j_k}{|Q_{j_1,\ldots,j_{k-1}}|n_0} \prod_{s=1}^{k-1} S\left( \frac{X_{(s)} - M(I_{j_1,\ldots,j_s})}{|I_{j_1,\ldots,j_s}|} \right).$$

Similarly, we have

$$\int_0^{X_{(k)}} \ddot{f}^k(X_{(1)}, \ldots, X_{(k-1)}, X)dX \geq \frac{N_0^{d-k}(j_k - 1)}{|Q_{j_1,\ldots,j_{k-1}}|n_0} \prod_{s=1}^{k-1} S\left( \frac{X_{(s)} - M(I_{j_1,\ldots,j_s})}{|I_{j_1,\ldots,j_s}|} \right).$$

Since

$$\ddot{f}^{k-1}(X_{(1)}, \ldots, X_{(k-1)}) = \frac{N_0^{d-k+1}}{|Q_{j_1,\ldots,j_{k-1}}|n_0} \prod_{s=1}^{k-1} S\left( \frac{X_{(s)} - M(I_{j_1,\ldots,j_s})}{|I_{j_1,\ldots,j_s}|} \right),$$

we have

$$\frac{j_k - 1}{N_0} \leq \ddot{F}_k(X_{(k)}|X_{(1)}, \ldots, X_{(k-1)}) < \frac{j_k}{N_0}$$

Thus, the first property is satisfied. It is easy to check the second property as $\ddot{f}(\vec{X})$ is a smooth function.

### D.2 PROOF OF THEOREM 1

**Proof for kernel density estimator**

Recall estimator can be written as

$$\hat{\mu}_{CT} = \frac{\sum_{i_1,i_2=1}^n Y_{i_1}(1 - Z_{i_1})K_h(\Phi(\vec{X}_{i_1}) - \Phi(\vec{X}_{i_2}))Z_{i_2}}{\sum_{i_1,i_2=1}^n (1 - Z_{i_1})K_h(\Phi(\vec{X}_{i_1}) - \Phi(\vec{X}_{i_2}))Z_{i_2}}.$$

In this proof, we write $\vec{U}_i = \Phi(\vec{X}_i)$. Then, we can define $P = \mathbb{P}(Z = 1)$,

$$\theta_U = P(1 - P) \int f_C^\Phi(\vec{U})\mu_C^\Phi(\vec{U})f_T^\Phi(\vec{U})d\vec{U} \qquad \text{and} \qquad \theta_L = P(1 - P) \int f_C^\Phi(\vec{U})f_T^\Phi(\vec{U})d\vec{U}$$

We can also define

$$T_U = \frac{1}{n(n-1)} \sum_{i_1,i_2=1}^n Y_{i_1}(1 - Z_{i_1})K_h(\vec{U}_{i_1} - \vec{U}_{i_2})Z_{i_2}$$

and

$$T_L = \frac{1}{n(n-1)} \sum_{i_1,i_2=1}^n (1 - Z_{i_1})K_h(\vec{U}_{i_1} - \vec{U}_{i_2})Z_{i_2}.$$

The proof is divided into three steps.

**Step 1.** In this step, we show that

$$\mathbb{E}(T_U - \theta_U)^2 \le C\left(\frac{1}{n^2 h^d} + \frac{1}{n} + h^{2(\alpha+\beta)}\right)$$

and

$$T_U - \theta_U = \frac{1}{n}\sum_{i=1}^{n}\left((1-P)\mu_C^\Phi(\vec{U}_i)f_C^\Phi(\vec{U}_i)Z_i + PY_i(1-Z_i)f_T^\Phi(\vec{U}_i) - 2\theta_U\right) + O_p\left(\frac{1}{nh^{d/2}} + h^{\alpha+\beta}\right).$$

$T_U$ is actually a U-statistic, so we decompose $T_U$ as

$$T_U - \theta_U$$
$$= \frac{1}{n(n-1)}\sum_{i_1\ne i_2}Y_{i_1}(1-Z_{i_1})K_h(\vec{U}_{i_1}-\vec{U}_{i_2})Z_{i_2} - \theta_U$$
$$= \frac{1}{n(n-1)}\sum_{i_1\ne i_2}\left(Y_{i_1}(1-Z_{i_1})K_h(\vec{U}_{i_1}-\vec{U}_{i_2})Z_{i_2} - (1-P)\int \mu_C^\Phi(\vec{U})f_C^\Phi(\vec{U})K_h(\vec{U}-\vec{U}_{i_2})Z_{i_2}d\vec{U}\right.$$
$$\left. - P\int Y_{i_1}(1-Z_{i_1})K_h(\vec{U}_{i_1}-\vec{U})f_T^\Phi(\vec{U})d\vec{U} + \tilde{\theta}_U\right)$$
$$+ \frac{1}{n}\sum_{i_2=1}^{n}\left((1-P)\int \mu_C^\Phi(\vec{U})f_C^\Phi(\vec{U})K_h(\vec{U}-\vec{U}_{i_2})Z_{i_2}d\vec{U} - \tilde{\theta}_U - (1-P)\mu_C^\Phi(\vec{U}_{i_2})f_C^\Phi(\vec{U}_{i_2})Z_{i_2} + \theta_U\right)$$
$$+ \frac{1}{n}\sum_{i_1=1}^{n}\left(P\int Y_{i_1}(1-Z_{i_1})K_h(\vec{U}_{i_1}-\vec{U})f_T^\Phi(\vec{U})d\vec{U} - \tilde{\theta}_U - PY_{i_1}(1-Z_{i_1})f_T^\Phi(\vec{U}_{i_1}) + \theta_U\right)$$
$$+ \frac{1}{n}\sum_{i_2=1}^{n}\left((1-P)\mu_C^\Phi(\vec{U}_{i_2})f_C^\Phi(\vec{U}_{i_2})Z_{i_2} - \theta_U\right)$$
$$+ \frac{1}{n}\sum_{i_1=1}^{n}\left(PY_{i_1}(1-Z_{i_1})f_T^\Phi(\vec{U}_{i_1}) - \theta_U\right)$$
$$+ \tilde{\theta}_U - \theta_U$$
$$= A_1 + A_2 + A_3 + A_4 + A_5 + A_6,$$

where $\tilde{\theta}_U = P(1-P)\int \mu_C^\Phi(\vec{U})f_C^\Phi(\vec{U})K_h(\vec{U}-\vec{U}')f_T^\Phi(\vec{U}')d\vec{U}d\vec{U}'$. We now bound each term. For $A_1$, we have

$$\mathbb{E}(A_1^2) \le \frac{C}{n(n-1)}\mathbb{E}(Y_{i_1}K_h(\vec{U}_{i_1}-\vec{U}_{i_2}))^2 \le \frac{C}{n(n-1)h^d}.$$

In terms of $A_2$, we have

$$\mathbb{E}(A_2^2) \le \frac{C}{n}\mathbb{E}\left(\int \mu_C^\Phi(\vec{U})K_h(\vec{U}-\vec{U}_{i_2})f_C^\Phi(\vec{U})d\vec{U} - \mu_C^\Phi(\vec{U}_{i_2})f_C^\Phi(\vec{U}_{i_2})\right)^2$$
$$\le \frac{C}{n}\mathbb{E}\left(\mu_C^\Phi f_C^\Phi \otimes K_h(\vec{U}_{i_2}) - \mu_C^\Phi(\vec{U}_{i_2})f_C^\Phi(\vec{U}_{i_2})\right)^2$$
$$\le \frac{Ch^{2\alpha}}{n}.$$

Here, we use the fact that $\alpha < \gamma$ and $\otimes$ represents convolution. With similar techniques, we have

$$
\begin{aligned}
\mathbb{E}(A_3^2) \leq & \frac{C}{n} \mathbb{E}\left(\int Y_{i_1} K_h(\vec{U}_{i_1} - \vec{U}) f_T^\Phi(\vec{U}) d\vec{U} - Y_{i_1} f_T^\Phi(\vec{U}_{i_1})\right)^2 \\
\leq & \frac{C}{n} \mathbb{E}\left(Y_{i_1}(f_T^\Phi \otimes K_h(\vec{U}_{i_1}) - f_T^\Phi(\vec{U}_{i_1}))\right)^2 \\
\leq & \frac{C}{n} \mathbb{E}(Y_{i_1}^2) \mathbb{E}(f_T^\Phi \otimes K_h(\vec{U}_{i_1}) - f_T^\Phi(\vec{U}_{i_1}))^2 \\
\leq & \frac{Ch^{2\beta}}{n}
\end{aligned}
$$

Applying the central limit theorem to $A_4$ and $A_5$ yields

$$
\begin{aligned}
\sqrt{n}(A_4 + A_5) = & \frac{1}{\sqrt{n}} \sum_{i=1}^n \left((1-P)\mu_C^\Phi(\vec{U}_i) f_C^\Phi(\vec{U}_i) Z_i + PY_i(1-Z_i) f_T^\Phi(\vec{U}_i) - 2\theta_U\right) \\
& \to N(0, V(\mu_C^\Phi, f_C^\Phi, f_T^\Phi)),
\end{aligned}
$$

where

$$
V(\mu_C^\Phi, f_C^\Phi, f_T^\Phi) = P^2(1-P)^2\left(\frac{\int \mu_C^{\Phi 2} f_C^{\Phi 2} f_T^\Phi}{P} + \frac{\int (\sigma^2 + \mu_C^{\Phi 2}) f_C^\Phi f_T^{\Phi 2}}{1-P} - 4\left(\int \mu_C^\Phi f_C^\Phi f_T^\Phi\right)^2\right).
$$

For the $A_6$, we apply the similar technique in Giné & Nickl (2008; 2016).

$$
\begin{aligned}
& \int \mu_C^\Phi(\vec{U}) f_C^\Phi(\vec{U}) K_h(\vec{U} - \vec{U}') f_T^\Phi(\vec{U}') d\vec{U} d\vec{U}' - \int \mu_C^\Phi(\vec{U}) f_C^\Phi(\vec{U}) f_T^\Phi(\vec{U}) d\vec{U} \\
= & \int \mu_C^\Phi(\vec{U}) f_C^\Phi(\vec{U}) K_h(\vec{U} - \vec{U}')[f_T^\Phi(\vec{U}') - f_T^\Phi(\vec{U})] d\vec{U} d\vec{U}' \\
= & \int \mu_C^\Phi(\vec{U}) f_C^\Phi(\vec{U}) K(\vec{V})[f_T^\Phi(\vec{U} - h\vec{V}) - f_T^\Phi(\vec{U})] d\vec{U} d\vec{V} \\
= & \int K(\vec{V})[\bar{f}_T^\Phi \otimes \mu_C^\Phi f_C^\Phi(h\vec{V}) - \bar{f}_T^\Phi \otimes \mu_C^\Phi f_C^\Phi(0)] d\vec{V}.
\end{aligned}
$$

Here, $\bar{f}_T^\Phi(\vec{x}) = f_T^\Phi(-\vec{x})$ and $\otimes$ represents convolution. Since $\mu_C^\Phi(\vec{x}) f_C^\Phi(\vec{x}) \in \mathcal{H}^\alpha([0,1]^d)$ and $\bar{f}_T^\Phi(\vec{x}) \in \mathcal{H}^\beta([0,1]^d)$, we can know $\bar{f}_T^\Phi \otimes \mu_C^\Phi f_C^\Phi(\vec{x}) \in \mathcal{H}^{\alpha+\beta}([0,1]^d)$. Then,

$$
|A_6| \leq h^{\alpha+\beta}.
$$

Putting $A_i, i = 1, \ldots, 6$ together yields the conclusion.

**Step 2.** In this step, we work on $T_L$ to obtain

$$
\mathbb{E}(T_L - \theta_L)^2 \leq C\left(\frac{1}{n^2 h^d} + \frac{1}{n} + h^{2(\beta+\gamma)}\right)
$$

and

$$
\mathbb{P}\left(|T_L - \tilde{\theta}_L| > C\eta\left(\sqrt{\frac{\log n}{n}} + \frac{\log n}{n h^{d/2}}\right)\right) \leq \frac{1}{n^\eta},
$$

where $\tilde{\theta}_L = P(1-P) \int f_C^\Phi(\vec{U}) K_h(\vec{U} - \vec{U}') f_T^\Phi(\vec{U}') d\vec{U} d\vec{U}'$ and $\eta$ is an arbitrary large constant. We decompose $T_L$ as

$$
\begin{aligned}
T_L - \theta_L =& \frac{1}{n(n-1)} \sum_{i_1 \neq i_2} (1 - Z_{i_1}) K_h(\vec{U}_{i_1} - \vec{U}_{i_2}) Z_{i_2} - \theta_L \\
=& \frac{1}{n(n-1)} \sum_{i_1 \neq i_2} \left( (1 - Z_{i_1}) K_h(\vec{U}_{i_1} - \vec{U}_{i_2}) Z_{i_2} - (1 - P) \int K_h(\vec{U} - \vec{U}_{i_2}) f_C^\Phi(\vec{U}) d\vec{U} Z_{i_2} \right. \\
& \left. - P \int (1 - Z_{i_1}) K_h(\vec{U}_{i_1} - \vec{U}) f_T^\Phi(\vec{U}) d\vec{U} + \tilde{\theta}_L \right) \\
&+ \frac{1}{n} \sum_{i_1=1}^n \left( P \int (1 - Z_{i_1}) K_h(\vec{U}_{i_1} - \vec{U}) f_T^\Phi(\vec{U}) d\vec{U} - P(1 - Z_{i_1}) f_T^\Phi(\vec{U}_{i_1}) \right) \\
&+ \frac{1}{n} \sum_{i_2=1}^n \left( (1 - P) \int K_h(\vec{U} - \vec{U}_{i_2}) f_C^\Phi(\vec{U}) d\vec{U} Z_{i_2} - (1 - P) Z_{i_2} f_C^\Phi(\vec{U}_{i_1}) \right) \\
&+ \frac{1}{n} \sum_{i=1}^n \left( (1 - P) Z_i f_C^\Phi(\vec{U}_i) + P(1 - Z_i) f_T(\vec{U}_i) - 2\tilde{\theta}_L \right) \\
&+ \tilde{\theta}_L - \theta_L
\end{aligned}
$$

where $\tilde{\theta}_L = P(1-P) \int f_C^\Phi(\vec{U}) K_h(\vec{U} - \vec{U}') f_T^\Phi(\vec{U}') d\vec{U} d\vec{U}'$. If we apply a similar proof in the last step, we can get

$$
\mathbb{E}(T_L - \theta_L)^2 \leq C \left( \frac{1}{n^2 h^d} + \frac{1}{n} + h^{2(\beta + \gamma)} \right).
$$

Since $T_L$ is a U-statistics, we can apply concentration inequality in Giné et al. (2000) (also see Lemma 9 in Shen et al., 2020). By following the notation of Lemma 9 in Shen et al. (2020), we have

$$
B_1 \leq C, \ B_2 \leq C\sqrt{n} h^{-d/2}, \ B_3 \leq Ch^{-d}, \ v_1^2 \leq C, \ v_2^2 \leq Ch^{-d},
$$

which leads to

$$
\mathbb{P}\left( |T_L - \tilde{\theta}_L| > C\eta \left( \sqrt{\frac{\log n}{n}} + \frac{\log n}{n h^{d/2}} \right) \right) \leq \frac{1}{n^\eta},
$$

where $\eta$ is an arbitrary large constant.

**Step 3.** In this step, we put the results of steps 1 and 2 together. Define the event

$$
\mathcal{A} = \left\{ |T_L - \tilde{\theta}_L| > C\eta \left( \sqrt{\frac{\log n}{n}} + \frac{\log n}{n h^{d/2}} \right) \right\}.
$$

Recall $\theta_U$ and $\theta_L$ are

$$
\theta_U = P(1-P) \int f_C^\Phi(\vec{U}) \mu_C^\Phi(\vec{U}) f_T^\Phi(\vec{U}) d\vec{U} \qquad \text{and} \qquad \theta_L = P(1-P) \int f_C^\Phi(\vec{U}) f_T^\Phi(\vec{U}) d\vec{U}.
$$

By variable transformation, we can rewrite them as

$$
\theta_U = P(1-P) \int \frac{f_C(\vec{X})}{\tilde{f}_C(\vec{X})} \mu_C(\vec{X}) f_T(\vec{X}) d\vec{X} \qquad \text{and} \qquad \theta_L = P(1-P) \int \frac{f_C(\vec{X})}{\tilde{f}_C(\vec{X})} f_T(\vec{X}) d\vec{X}.
$$

If we define $\theta_U^0$ and $\theta_L^0$ as

$$
\theta_U^0 = P(1-P)\mu_{CT} \qquad \text{and} \qquad \theta_L^0 = P(1-P),
$$

then we can know that

$$
(\theta_U^0 - \theta_U)^2, (\theta_L^0 - \theta_L)^2 \leq C \|\tilde{f}_C(\vec{X}) - f_C(\vec{X})\|_2^2.
$$

This immediately suggests that

$$
\mathbb{E}(T_U - \theta_U^0)^2 \leq C \left( \frac{1}{n^2 h^d} + \frac{1}{n} + h^{2(\alpha + \beta)} + \|\tilde{f}_C(\vec{X}) - f_C(\vec{X})\|_2^2 \right)
$$

and

$$\mathbb{E}(T_L - \theta_L^0)^2 \leq C \left( \frac{1}{n^2 h^d} + \frac{1}{n} + h^{2(\beta+\gamma)} + \|\tilde{f}_C(\vec{X}) - f_C(\vec{X})\|_2^2 \right).$$

Then, we have

$$\mathbb{E}(\hat{\mu}_{CT} - \mu_{CT})^2 = \mathbb{E} \left( \frac{T_U}{T_L} - \mu_{CT} \right)^2$$

$$= \mathbb{E} \left( \left( \frac{T_U}{T_L} - \mu_{CT} \right)^2 \mathbb{I}_{\mathcal{A}} \right) + \mathbb{E} \left( \left( \frac{T_U}{T_L} - \mu_{CT} \right)^2 \mathbb{I}_{\mathcal{A}^c} \right).$$

We work on above two terms separately. For the first one, when $n$ is large enough, we have

$$T_L \geq \frac{\theta_L^0}{\sqrt{2}} \qquad \text{on event } \mathcal{A}.$$

Thus,

$$\mathbb{E} \left( \left( \frac{T_U}{T_L} - \mu_{CT} \right)^2 \mathbb{I}_{\mathcal{A}} \right) \leq \frac{2}{\theta_L^{0\,2}} \mathbb{E}(T_U - T_L \mu_{CT})^2$$

$$\leq \frac{2}{\theta_L^{0\,2}} \left( \mathbb{E}(T_U - \theta_U^0)^2 + \mu_{CT}^2 \mathbb{E}(T_L - \theta_L^0)^2 \right)$$

$$\leq C \left( \frac{1}{n^2 h^d} + \frac{1}{n} + h^{2(\alpha+\beta)} + \|\tilde{f}_C(\vec{X}) - f_C(\vec{X})\|_2^2 \right)$$

$$\leq C \left( n^{-\frac{4(\alpha+\beta)}{d+2(\alpha+\beta)}} + n^{-1} + \|\tilde{f}_C(\vec{X}) - f_C(\vec{X})\|_2^2 \right).$$

The last step is due to $h = n^{-\frac{2}{d+2(\alpha+\beta)}}$. For the second term, we have

$$\mathbb{E} \left( \left( \frac{T_U}{T_L} - \mu_{CT} \right)^2 \mathbb{I}_{\mathcal{A}^c} \right) \leq 2\mathbb{E}(\hat{\mu}_{CT}^2 \mathbb{I}_{\mathcal{A}^c}) + 2\mu_{CT}^2 \mathbb{E}(\mathbb{I}_{\mathcal{A}^c})$$

$$\leq 2\mathbb{E}(\max_{i:Z_i=0} Y_i^2 \mathbb{I}_{\mathcal{A}^c}) + 2\mu_{CT}^2 \mathbb{P}(\mathcal{A}^c)$$

$$\leq 4\mathbb{E}(\max_{i:Z_i=0} \epsilon_i^2 \mathbb{I}_{\mathcal{A}^c}) + (2\mu_{CT}^2 + 4\sup_{\vec{X}} \mu_C^2(\vec{X})) \mathbb{P}(\mathcal{A}^c)$$

$$\leq (4\mathbb{E}(\max_{i:Z_i=0} \epsilon_i^2) + 2\mu_{CT}^2 + 4\sup_{\vec{X}} \mu_C^2(\vec{X})) \mathbb{P}(\mathcal{A}^c)$$

$$\leq Cn^{1-\eta}$$

Here, we use the fact that

$$\mathbb{E}(\max_{i:Z_i=0} \epsilon_i^2) = \int_0^\infty \mathbb{P}(\max_{i:Z_i=0} \epsilon_i^2 > t)dt \leq n \int_0^\infty \mathbb{P}(\epsilon_i^2 > t)dt = n\mathbb{E}(\epsilon_i^2).$$

If we choose $\eta > 2$, we can conclude that

$$\mathbb{E}(\hat{\mu}_{CT} - \mu_{CT})^2 \leq C \left( n^{-\frac{4(\alpha+\beta)}{d+2(\alpha+\beta)}} + n^{-1} + \|\tilde{f}_C(\vec{X}) - f_C(\vec{X})\|_2^2 \right).$$

**Proof for projection density estimator**

Similar with the proof for kernel density estimator, we define

$$T_U = \frac{1}{n(n-1)} \sum_{i_1,i_2=1}^n Y_{i_1}(1 - Z_{i_1}) K_L(\vec{U}_{i_1}, \vec{U}_{i_2}) Z_{i_2}$$

and

$$T_L = \frac{1}{n(n-1)} \sum_{i_1,i_2=1}^n (1 - Z_{i_1}) K_L(\vec{U}_{i_1}, \vec{U}_{i_2}) Z_{i_2}.$$

Similarly, we use the same notation of $P$, $\theta_U$ and $\theta_L$.

**Step 1.** In this step, we show that

$$\mathbb{E}(T_U - \theta_U)^2 \leq C \left( \frac{L}{n^2} + \frac{1}{n} + L^{-\frac{2(\alpha+\beta)}{d}} \right)$$

and

$$T_U - \theta_U = \frac{1}{n} \sum_{i=1}^n \left( (1-P)\mu_C^\Phi(\vec{U}_i)f_C^\Phi(\vec{U}_i)Z_i + PY_i(1-Z_i)f_T^\Phi(\vec{U}_i) - 2\theta_U \right) + O_p \left( \frac{L}{n^2} + L^{-\frac{2(\alpha+\beta)}{d}} \right).$$

$T_U$ is actually a U-statistics, so we decompose $T_U$ as

$$
\begin{aligned}
& T_U - \theta_U \\
={}& \frac{1}{n(n-1)} \sum_{i_1 \neq i_2} Y_{i_1}(1-Z_{i_1})K_L(\vec{U}_{i_1}, \vec{U}_{i_2})Z_{i_2} - \theta_U \\
={}& \frac{1}{n(n-1)} \sum_{i_1 \neq i_2} \left( Y_{i_1}(1-Z_{i_1})K_L(\vec{U}_{i_1}, \vec{U}_{i_2})Z_{i_2} - (1-P)\int \mu_C^\Phi(\vec{U})f_C^\Phi(\vec{U})K_L(\vec{U}, \vec{U}_{i_2})Z_{i_2}d\vec{U} \right. \\
& \left. - P \int Y_{i_1}(1-Z_{i_1})K_L(\vec{U}_{i_1}, \vec{U})f_T^\Phi(\vec{U})d\vec{U} + \tilde{\theta}_U \right) \\
& + \frac{1}{n} \sum_{i_2=1}^n \left( (1-P)\int \mu_C^\Phi(\vec{U})f_C^\Phi(\vec{U})K_L(\vec{U}, \vec{U}_{i_2})Z_{i_2}d\vec{U} - \tilde{\theta}_U - (1-P)\mu_C^\Phi(\vec{U}_{i_2})f_C^\Phi(\vec{U}_{i_2})Z_{i_2} + \theta_U \right) \\
& + \frac{1}{n} \sum_{i_1=1}^n \left( P \int Y_{i_1}(1-Z_{i_1})K_L(\vec{U}_{i_1}, \vec{U})f_T^\Phi(\vec{U})d\vec{U} - \tilde{\theta}_U - PY_{i_1}(1-Z_{i_1})f_T^\Phi(\vec{U}_{i_1}) + \theta_U \right) \\
& + \frac{1}{n} \sum_{i_2=1}^n \left( (1-P)\mu_C^\Phi(\vec{U}_{i_2})f_C^\Phi(\vec{U}_{i_2})Z_{i_2} - \theta_U \right) \\
& + \frac{1}{n} \sum_{i_1=1}^n \left( PY_{i_1}(1-Z_{i_1})f_T^\Phi(\vec{U}_{i_1}) - \theta_U \right) \\
& + \tilde{\theta}_U - \theta_U \\
={}& A_1 + A_2 + A_3 + A_4 + A_5 + A_6,
\end{aligned}
$$

where $\tilde{\theta}_U = P(1-P) \int \mu_C^\Phi(\vec{U})f_C^\Phi(\vec{U})K_L(\vec{U}, \vec{U}')f_T^\Phi(\vec{U}')d\vec{U}d\vec{U}'$. We now bound each term. For $A_1$, we have

$$\mathbb{E}(A_1^2) \leq \frac{C}{n(n-1)} \mathbb{E}(Y_{i_1}K_L(\vec{U}_{i_1}, \vec{U}_{i_2}))^2 \leq \frac{C}{n(n-1)} \int K_L(\vec{U}_{i_1}, \vec{U}_{i_2})^2 d\vec{U}_{i_1}d\vec{U}_{i_2} \leq \frac{CL}{n(n-1)}.$$

Since $\psi_l$ is the basis function, we can write

$$\mu_C^\Phi(\vec{U})f_C^\Phi(\vec{U}) = \sum_{l=1}^\infty a_l \psi_l(\vec{U}) \qquad \text{and} \qquad f_T^\Phi(\vec{U}) = \sum_{l=1}^\infty b_l \psi_l(\vec{U}),$$

where

$$|a_l| \leq l^{-(\alpha/d+1/2)}, \qquad |b_l| \leq l^{-(\beta/d+1/2)}.$$

Here, we use the fact that $\alpha < \gamma$. In terms of $A_2$, we have

$$
\begin{aligned}
\mathbb{E}(A_2^2) &\leq \frac{C}{n}\mathbb{E}\left(\int \mu_C^\Phi(\vec{U})K_L(\vec{U},\vec{U}_{i_2})f_C^\Phi(\vec{U})d\vec{U} - \mu_C^\Phi(\vec{U}_{i_2})f_C^\Phi(\vec{U}_{i_2})\right)^2 \\
&\leq \frac{C}{n}\mathbb{E}\left(\sum_{l=1}^{L} a_l\psi_l(\vec{U}_{i_2}) - \mu_C^\Phi(\vec{U}_{i_2})f_C^\Phi(\vec{U}_{i_2})\right)^2 \\
&\leq \frac{C}{n}\mathbb{E}\left(\sum_{l=L+1}^{\infty} a_l\psi_l(\vec{U}_{i_2})\right)^2 \\
&\leq \frac{CL^{-2\alpha/d}}{n}.
\end{aligned}
$$

Similarly, we have

$$
\begin{aligned}
\mathbb{E}(A_3^2) &\leq \frac{C}{n}\mathbb{E}\left(\int Y_{i_1}K_L(\vec{U}_{i_1},\vec{U})f_T^\Phi(\vec{U})d\vec{U} - Y_{i_1}f_T^\Phi(\vec{U}_{i_1})\right)^2 \\
&\leq \frac{C}{n}\mathbb{E}\left(Y_{i_1}\left(\sum_{l=1}^{L} b_l\psi_l(\vec{U}_{i_1}) - f_T^\Phi(\vec{U}_{i_1})\right)\right)^2 \\
&\leq \frac{C}{n}\mathbb{E}(Y_{i_1}^2)L^{-2\beta/d} \\
&\leq \frac{CL^{-2\beta/d}}{n}
\end{aligned}
$$

With the same argument in the proof of kernel density estimator, we have

$$
\begin{aligned}
\sqrt{n}(A_4 + A_5) &= \frac{1}{\sqrt{n}}\sum_{i=1}^{n}\left((1-P)\mu_C^\Phi(\vec{U}_i)f_C^\Phi(\vec{U}_i)Z_i + PY_i(1-Z_i)f_T^\Phi(\vec{U}_i) - 2\theta_U\right) \\
&\to N(0, V(\mu_C^\Phi, f_C^\Phi, f_T^\Phi)).
\end{aligned}
$$

For the $A_6$, we have

$$
\begin{aligned}
&\int \mu_C^\Phi(\vec{U})f_C^\Phi(\vec{U})K_L(\vec{U},\vec{U}')f_T^\Phi(\vec{U}')d\vec{U}d\vec{U}' - \int \mu_C^\Phi(\vec{U})f_C^\Phi(\vec{U})f_T^\Phi(\vec{U})d\vec{U} \\
&= \sum_{l=1}^{L} a_l b_l - \sum_{l=1}^{\infty} a_l b_l \\
&= -\sum_{l=L+1}^{\infty} a_l b_l.
\end{aligned}
$$

Because $|a_l b_l| \leq l^{-(\alpha+\beta)/d}$,

$$
|A_6| \leq L^{-(\alpha+\beta)/d}.
$$

Putting $A_i, i = 1, \ldots, 6$ together yields the conclusion.

**Step 2.** In this step, we work on $T_L$ to obtain

$$
\mathbb{E}(T_L - \theta_L)^2 \leq C\left(\frac{L}{n^2} + \frac{1}{n} + L^{-2(\beta+\gamma)/d}\right)
$$

and

$$
\mathbb{P}\left(|T_L - \tilde{\theta}_L| > C\eta\left(\sqrt{\frac{\log n}{n}} + \frac{\log n\sqrt{L}}{n}\right)\right) \leq \frac{1}{n^\eta},
$$

where $\tilde{\theta}_L = P(1-P) \int f_C^{\Phi}(\vec{U}) K_L(\vec{U}, \vec{U}') f_T^{\Phi}(\vec{U}') d\vec{U} d\vec{U}'$ and $\eta$ is an arbitrary large constant. We decompose $T_L$ as

$$
\begin{aligned}
T_L - \theta_L = & \frac{1}{n(n-1)} \sum_{i_1 \neq i_2} (1 - Z_{i_1}) K_L(\vec{U}_{i_1}, \vec{U}_{i_2}) Z_{i_2} - \theta_L \\
= & \frac{1}{n(n-1)} \sum_{i_1 \neq i_2} \left( (1 - Z_{i_1}) K_L(\vec{U}_{i_1}, \vec{U}_{i_2}) Z_{i_2} - (1 - P) \int K_L(\vec{U}, \vec{U}_{i_2}) f_C^{\Phi}(\vec{U}) d\vec{U} Z_{i_2} \right. \\
& \left. - P \int (1 - Z_{i_1}) K_L(\vec{U}_{i_1}, \vec{U}) f_T^{\Phi}(\vec{U}) d\vec{U} + \tilde{\theta}_L \right) \\
& + \frac{1}{n} \sum_{i_1=1}^{n} \left( P \int (1 - Z_{i_1}) K_L(\vec{U}_{i_1}, \vec{U}) f_T^{\Phi}(\vec{U}) d\vec{U} - P(1 - Z_{i_1}) f_T^{\Phi}(\vec{U}_{i_1}) \right) \\
& + \frac{1}{n} \sum_{i_2=1}^{n} \left( (1 - P) \int K_L(\vec{U}, \vec{U}_{i_2}) f_C^{\Phi}(\vec{U}) d\vec{U} Z_{i_2} - (1 - P) Z_{i_2} f_C^{\Phi}(\vec{U}_{i_1}) \right) \\
& + \frac{1}{n} \sum_{i=1}^{n} \left( (1 - P) Z_i f_C^{\Phi}(\vec{U}_i) + P(1 - Z_i) f_T(\vec{U}_i) - 2\tilde{\theta}_L \right) \\
& + \tilde{\theta}_L - \theta_L
\end{aligned}
$$

where $\tilde{\theta}_L = P(1-P) \int f_C^{\Phi}(\vec{U}) K_L(\vec{U}, \vec{U}') f_T^{\Phi}(\vec{U}') d\vec{U} d\vec{U}'$. If we apply a similar proof in the last step, we can get

$$
\mathbb{E}(T_L - \theta_L)^2 \leq C \left( \frac{L}{n^2} + \frac{1}{n} + L^{-2(\beta+\gamma)/d} \right).
$$

Since $T_L$ is a U-statistics, we can apply concentration inequality in Giné et al. (2000) (also see Lemma 9 in Shen et al., 2020). By following the notation of Lemma 9 in Shen et al. (2020), we have

$$
B_1 \leq C, \ B_2 \leq C\sqrt{n}L^{1/2}, \ B_3 \leq CL^{1/2}, \ v_1^2 \leq C, \ v_2^2 \leq CL,
$$

which leads to

$$
\mathbb{P}\left( |T_L - \tilde{\theta}_L| > C\eta \left( \sqrt{\frac{\log n}{n}} + \frac{\log n \sqrt{L}}{n} \right) \right) \leq \frac{1}{n^{\eta}},
$$

where $\eta$ is an arbitrary large constant.

**Step 3.** In this step, we put the results of steps 1 and 2 together. Recall $\theta_U$ and $\theta_L$ are

$$
\theta_U = P(1-P) \int \frac{f_C(\vec{X})}{\tilde{f}_C(\vec{X})} \mu_C(\vec{X}) f_T(\vec{X}) d\vec{X} \qquad \text{and} \qquad \theta_L = P(1-P) \int \frac{f_C(\vec{X})}{\tilde{f}_C(\vec{X})} f_T(\vec{X}) d\vec{X}.
$$

If we define $\theta_U^0$ and $\theta_L^0$ as

$$
\theta_U^0 = P(1-P)\mu_{CT} \qquad \text{and} \qquad \theta_L^0 = P(1-P),
$$

then we can know that

$$
\mathbb{E}(T_U - \theta_U^0)^2 \leq C \left( \frac{L}{n^2} + \frac{1}{n} + L^{-2(\alpha+\beta)/d} + \|\tilde{f}_C(\vec{X}) - f_C(\vec{X})\|_2^2 \right)
$$

and

$$
\mathbb{E}(T_L - \theta_L^0)^2 \leq C \left( \frac{L}{n^2} + \frac{1}{n} + L^{-2(\beta+\gamma)/d} + \|\tilde{f}_C(\vec{X}) - f_C(\vec{X})\|_2^2 \right).
$$

As $L = n^{\frac{2d}{d+2(\alpha+\beta)}}$, an application of the same argument in the proof of kernel estimator yields

$$
\mathbb{E}(\hat{\mu}_{CT} - \mu_{CT})^2 \leq C \left( n^{-\frac{4(\alpha+\beta)}{d+2(\alpha+\beta)}} + n^{-1} + \|\tilde{f}_C(\vec{X}) - f_C(\vec{X})\|_2^2 \right).
$$

### D.3 PROOF OF COROLLARY 1

We follow the notation in the proof of Theorem 1. Now, we prove the first part of the corollary. By Theorem 1, we have

$$\mathbb{E}\left((\hat{\mu}_{CT} - \mu_{CT})^2 \Big| \tilde{f}_C(\vec{X})\right) \leq C\left(n^{-\frac{4(\alpha+\beta)}{d+2(\alpha+\beta)}} + n^{-1} + \|\tilde{f}_C(\vec{X}) - f_C(\vec{X})\|_2^2\right).$$

Due to $\alpha + \beta \leq d/2$, we have

$$\mathbb{E}(\hat{\mu}_{CT} - \mu_{CT})^2 \leq C\left(n^{-\frac{4(\alpha+\beta)}{d+2(\alpha+\beta)}} + N^{-\frac{2\kappa}{d+2\kappa}}\right).$$

If we apply $\kappa > 2(\alpha + \beta)d/(td + (2t-4)(\alpha + \beta))$, we can get

$$\mathbb{E}(\hat{\mu}_{CT} - \mu_{CT})^2 \leq Cn^{-\frac{4(\alpha+\beta)}{d+2(\alpha+\beta)}}.$$

We turn to the second part of the corollary now. By the Step 1 in the proof of Theorem 1,

$$T_U - \theta_U = \frac{1}{n}\sum_{i=1}^{n}\left((1-P)\mu_C^\Phi(\vec{U}_i)f_C^\Phi(\vec{U}_i)Z_i + PY_i(1-Z_i)f_T^\Phi(\vec{U}_i) - 2\theta_U\right) + O_p\left(\frac{1}{nh^{d/2}} + h^{\alpha+\beta}\right).$$

By $\alpha + \beta \geq d/2$, the choice of $h$ suggests that

$$T_U - \theta_U = \frac{1}{n}\sum_{i=1}^{n}\left((1-P)\mu_C^\Phi(\vec{U}_i)f_C^\Phi(\vec{U}_i)Z_i + PY_i(1-Z_i)f_T^\Phi(\vec{U}_i) - 2\theta_U\right) + o_p\left(n^{-1/2}\right).$$

Because of $\kappa > d/2(t-1)$, we can know that

$$\theta_U^0 - \theta_U = o_p\left(n^{-1/2}\right).$$

Thus, we can conclude that

$$\frac{\sqrt{n}(T_U - \theta_U^0)}{\sqrt{V(\mu_C^\Phi, f_C^\Phi, f_T^\Phi)}} \to N(0,1),$$

where

$$V(\mu_C^\Phi, f_C^\Phi, f_T^\Phi) = P^2(1-P)^2\left(\frac{\int \mu_C^{\Phi 2}f_C^{\Phi 2}f_T^\Phi}{P} + \frac{\int(\sigma^2 + \mu_C^{\Phi 2})f_C^\Phi f_T^{\Phi 2}}{1-P} - 4\left(\int \mu_C^\Phi f_C^\Phi f_T^\Phi\right)^2\right).$$

The variable transformation suggests that

$$\frac{V(\mu_C^\Phi, f_C^\Phi, f_T^\Phi)}{P^2(1-P)^2}$$

$$= \frac{1}{P}\int \frac{\mu_C^2(\vec{x})f_C^2(\vec{x})f_T(\vec{x})}{\tilde{f}_C^2(\vec{x})}d\vec{x} + \frac{1}{1-P}\int \frac{(\sigma^2(\vec{x}) + \mu_C^2(\vec{x}))f_C(\vec{x})f_T^2(\vec{x})}{\tilde{f}_C^2(\vec{x})}d\vec{x}$$

$$- 4\left(\int \frac{\mu_C(\vec{x})f_C(\vec{x})f_T(\vec{x})}{\tilde{f}_C(\vec{x})}d\vec{x}\right)^2$$

$$\to \frac{1}{P}\int \mu_C^2(\vec{x})f_T(\vec{x})d\vec{x} + \frac{1}{1-P}\int \frac{(\sigma^2(\vec{x}) + \mu_C^2(\vec{x}))f_T^2(\vec{x})}{f_C(\vec{x})}d\vec{x} - 4\left(\int \mu_C(\vec{x})f_T(\vec{x})d\vec{x}\right)^2$$

$$= V$$

This leads to

$$\frac{\sqrt{n}(T_U - \theta_U^0)}{\sqrt{V}P(1-P)} \to N(0,1).$$

The proof of Theorem 1 also suggests that

$$T_L \to_p \theta_L^0 = P(1-P).$$

Therefore, we can conclude that

$$\frac{\sqrt{n}(\hat{\mu}_{CT} - \mu_{CT})}{\sqrt{V}} \to N(0,1).$$

### D.4 Proof of Theorem 2

Without loss of generality, we can assume $f_C(\vec{X})$ is uniform distribution and $\Phi$ is identity transformation. We prove the two parts of the lower bound by considering the two cases. **Case I:** In order to construct the worst case, we consider the following model where $Y$ only take values from $\{0, 1\}$, i.e., $Y$ is a binary variable. The distribution of this model can be written as

$$
\begin{aligned}
&\mathbb{P}(\vec{X}, Z, Y) \\
=&\mathbb{P}(Z)\mathbb{P}(\vec{X}|Z)\mathbb{P}(Y|\vec{X}, Z) \\
=&\left(Pf_T(\vec{X})\mu_T(\vec{X})^Y(1 - \mu_T(\vec{X}))^{1-Y}\right)^Z \left((1 - P)f_C(\vec{X})\mu_C(\vec{X})^Y(1 - \mu_C(\vec{X}))^{1-Y}\right)^{1-Z},
\end{aligned}
$$

where $P = \mathbb{P}(Z = 1)$. Let $L$ be the integer closest to $n^{2d/(d+2(\alpha+\beta))}$ and $H$ be a smooth function defined on $[0, 1]^d$ such that $\int H(\vec{x})d\vec{x} = 0$ and $\int H^2(\vec{x})d\vec{x} = 1$. The space $[0, 1]^d$ can be divided into $L$ non-overlap cubes of size $L^{-1/d} \times \ldots \times L^{-1/d}$. We name these small cubes $Q_1, \ldots, Q_L$. On each cube $Q_l$, we can insert a scaled and shifted version of $H$, i.e., define the following function

$$
\psi_l(\vec{x}) = H\left((\vec{x} - M(Q_l))L^{1/d}\right),
$$

where $M(\cdot)$ is the bottom left point of the cube. Clearly, we know $\psi_l$ is supported on $Q_l$ and have $\int \psi_l = 0$ and $\int \psi_l^2 = 1/L$. Now, we construct the least favorable hypothesis separately when $\alpha < \beta$ and $\alpha \geq \beta$.

$\alpha < \beta$ The hypothesis we consider are

$$
\mathbb{H}_0(\eta) : P = 1/2; f_C = 1, f_T = 1 + \sum_l \eta_l r_f \psi_l; \mu_C = 1/2, \mu_T = 1/2
$$

$$
\mathbb{H}_1(\eta) : P = 1/2; f_C = 1, f_T = 1 + \sum_l \eta_l r_f \psi_l; \mu_C = 1/2 + \sum_l \eta_l r_\mu \psi_l, \mu_T = 1/2.
$$

Here, $\eta = (\eta_1, \ldots, \eta_l, \ldots, \eta_L)$ is a sequence taking value from $\{-1, 1\}^L$, $r_f = L^{-\beta/d}$ and $r_\mu = L^{-\alpha/d}$. By construction, we can know that $f_T \in \mathcal{H}^\beta([0, 1]^d)$ and $\mu_C \in \mathcal{H}^\alpha([0, 1]^d)$. Let $\mathbb{P}_0(\eta)$ be distribution of $(\vec{X}, Z, Y)$ under $\mathbb{H}_0(\eta)$ and $\mathbb{P}_1(\eta)$ be distribution of $(\vec{X}, Z, Y)$ under $\mathbb{H}_1(\eta)$. The null and alternative hypothesis we consider here are mixture distributions $\mathbb{P}_0 = 2^{-L} \sum_{\eta \in \{-1, 1\}^L} \mathbb{P}_0(\eta)$ and $\mathbb{P}_1 = 2^{-L} \sum_{\eta \in \{-1, 1\}^L} \mathbb{P}_1(\eta)$. Simple calculation suggests that

$$
\mu_{CT} = 1/2 \quad \text{under } \mathbb{H}_0(\eta) \quad \text{and} \quad \mu_{CT} = 1/2 + n^{-\frac{2(\alpha+\beta)}{d+2(\alpha+\beta)}} \quad \text{under } \mathbb{H}_1(\eta).
$$

In order to calculate Hellinger distance between $\mathbb{P}_0$ and $\mathbb{P}_1$, we adopt Theorem 2.1 in Robins et al. (2009) to obtain

$$
H(\mathbb{P}_0^n, \mathbb{P}_1^n) \leq Cn^2 \frac{1}{L}(r_\mu^4 + r_\mu^2 r_f^2) \leq C.
$$

By applying Theorem 2.15 in Tsybakov (2008), we have

$$
\inf_{\hat{\mu}_{CT}} \sup_{F \in \mathcal{F}_{\alpha, \beta}} \mathbb{E}(\hat{\mu}_{CT} - \mu_{CT})^2 \geq cn^{-\frac{4(\alpha+\beta)}{d+2(\alpha+\beta)}}.
$$

$\alpha \geq \beta$ Now we consider the hypothesis

$$
\mathbb{H}_0(\eta) : P = 1/2; f_C = 1, f_T = 1; \mu_C = 1/2 + \sum_l \eta_l r_\mu \psi_l, \mu_T = 1/2
$$

$$
\mathbb{H}_1(\eta) : P = 1/2; f_C = 1, f_T = 1 + \sum_l \eta_l r_f \psi_l; \mu_C = 1/2 + \sum_l \eta_l r_\mu \psi_l, \mu_T = 1/2.
$$

We use the same notation as when $\alpha < \beta$. We can know that

$$
\mu_{CT} = 1/2 \quad \text{under } \mathbb{H}_0(\eta) \quad \text{and} \quad \mu_{CT} = 1/2 + n^{-\frac{2(\alpha+\beta)}{d+2(\alpha+\beta)}} \quad \text{under } \mathbb{H}_1(\eta).
$$

and

$$
H(\mathbb{P}_0^n, \mathbb{P}_1^n) \leq Cn^2 \frac{1}{L}(r_f^4 + r_\mu^2 r_f^2) \leq C.
$$

So we can conclude that

$$\inf_{\hat{\mu}_{CT}} \sup_{F \in \mathcal{F}_{\alpha,\beta}} \mathbb{E}(\hat{\mu}_{CT} - \mu_{CT})^2 \geq cn^{-\frac{4(\alpha+\beta)}{d+2(\alpha+\beta)}}.$$

**Case II:** In this case, we consider the same model in the previous case and the following hypothesis

$$\mathbb{H}_0 : P = 1/2; f_C = 1, f_T = 1; \mu_C = 1/2, \mu_T = 1/2$$
$$\mathbb{H}_1 : P = 1/2; f_C = 1, f_T = 1; \mu_C = 1/2 + r, \mu_T = 1/2.$$

Here, $r = n^{-1/2}$. $\mathbb{P}_0$ is distribution of $(\vec{X}, Z, Y)$ under $\mathbb{H}_0$ and $\mathbb{P}_1$ is distribution of $(\vec{X}, Z, Y)$ under $\mathbb{H}_1$. Calculation suggests that

$$\mu_{CT} = 1/2 \quad \text{under } \mathbb{H}_0 \quad \text{and} \quad \mu_{CT} = 1/2 + n^{-1/2} \quad \text{under } \mathbb{H}_1.$$

and

$$\int \frac{(d\mathbb{P}_1^n)^2}{d\mathbb{P}_0^n} = \left(1 + r^2\right)^n.$$

This implies

$$\chi^2(\mathbb{P}_1^n, \mathbb{P}_0^n) = \left(1 + r^2\right)^n - 1 \leq \exp\left(nr^2\right) - 1 \leq C.$$

By applying Theorem 2.2 in Tsybakov (2008), we have

$$\inf_{\hat{\mu}_{CT}} \sup_{F \in \mathcal{F}_{\alpha,\beta}} \mathbb{E}(\hat{\mu}_{CT} - \mu_{CT})^2 \geq cn^{-1}.$$

### D.5 PROOF OF THEOREM 3

**Proof for kernel density estimator** Through the proof , we define $\mathcal{A} := \left\{Z_i, \vec{X}_i(1 - Z_i)\right\}_{i=1,\ldots,n}$ and conduct the proof conditioned on $\mathcal{A}$. Conditioned on $\mathcal{A}$, the map $\hat{\Phi}_D$ can be seen as fixed. Recall

$$\hat{\mu}_{CT} = \frac{\sum_{i_1,i_2=1}^n Y_{i_1}(1 - Z_{i_1})K_h(\hat{\Phi}_D(\vec{X}_{i_1}) - \hat{\Phi}_D(\vec{X}_{i_2}))Z_{i_2}}{\sum_{i_1,i_2=1}^n (1 - Z_{i_1})K_h(\hat{\Phi}_D(\vec{X}_{i_1}) - \hat{\Phi}_D(\vec{X}_{i_2}))Z_{i_2}}.$$

We write $\theta_U = \mu_{CT}$, $\theta_L = 1$,

$$T_U = \frac{1}{n_0 n_1} \sum_{i_1,i_2=1}^n Y_{i_1}(1 - Z_{i_1})K_h(\hat{\Phi}_D(\vec{X}_{i_1}) - \hat{\Phi}_D(\vec{X}_{i_2}))Z_{i_2}$$

and

$$T_L = \frac{1}{n_0 n_1} \sum_{i_1,i_2=1}^n (1 - Z_{i_1})K_h(\hat{\Phi}_D(\vec{X}_{i_1}) - \hat{\Phi}_D(\vec{X}_{i_2}))Z_{i_2}.$$

The proof is divided into three steps.

**Step 1.** In this step, we show that

$$\mathbb{E}((T_U - \theta_U)^2 | \mathcal{A}) \leq C \left(\frac{1}{n_0 n_1 h^d} + \frac{1}{n_0} + \frac{1}{n_1} + \frac{1}{n_0^{2\alpha_0/d}} + h^{2(\alpha+\beta)}\right),$$

where $\alpha_0 = \alpha \wedge 1$. Denote $\vec{U}_i = \hat{\Phi}_D(\vec{X}_i)$ for $i = 1, \ldots, n$. We decompose the $T_U$ into two parts

$$T_U = \frac{1}{n_0 n_1} \sum_{i_1,i_2=1}^n Y_{i_1}(1 - Z_{i_1})K_h(\hat{\Phi}_D(\vec{X}_{i_1}) - \hat{\Phi}_D(\vec{X}_{i_2}))Z_{i_2}$$

$$= \frac{1}{n_0 n_1} \sum_{i_1:Z_{i_1}=0, i_2:Z_{i_2}=1} \epsilon_{i_1} K_h(\vec{U}_{i_1} - \vec{U}_{i_2}) + \frac{1}{n_0 n_1} \sum_{i_1:Z_{i_1}=0, i_2:Z_{i_2}=1} \mu_C^{\hat{\Phi}_D}(\vec{U}_{i_1})K_h(\vec{U}_{i_1} - \vec{U}_{i_2}),$$

where $\epsilon_i = Y_i - \mu_C^{\hat{\Phi}_D}(\vec{U}_i)$ when $Z_i = 0$. We work these two terms separately. The first term can be written as

$$\frac{1}{n_0 n_1} \sum_{i_1:Z_{i_1}=0, i_2:Z_{i_2}=1} \epsilon_{i_1} K_h(\vec{U}_{i_1} - \vec{U}_{i_2})$$

$$= \frac{1}{n_0 n_1} \sum_{i_1:Z_{i_1}=0, i_2:Z_{i_2}=1} \left( \epsilon_{i_1} K_h(\vec{U}_{i_1} - \vec{U}_{i_2}) - \epsilon_{i_1} \int K_h(\vec{U}_{i_1} - \vec{U}) f_T^{\hat{\Phi}_D}(\vec{U}) d\vec{U} \right)$$

$$+ \frac{1}{n_0} \sum_{i_1:Z_{i_1}=0} \epsilon_{i_1} \int K_h(\vec{U}_{i_1} - \vec{U}) f_T^{\hat{\Phi}_D}(\vec{U}) d\vec{U}$$

$$:= A_1 + A_2$$

In terms of $A_1$, we have

$$\mathbb{E}(A_1^2|\mathcal{A}) \le \mathbb{E}\left( \frac{4}{n_0^2 n_1^2} \sum_{i_1:Z_{i_1}=0, i_2:Z_{i_2}=1} \epsilon_{i_1}^2 K_h(\vec{U}_{i_1} - \vec{U}_{i_2})^2 \middle| \mathcal{A} \right)$$

$$\le \frac{C}{n_0 n_1 h^d}$$

For the $A_2$, note that $\epsilon_{i_1}$ are independent given $\mathcal{A}$, which yields

$$\mathbb{E}(A_2^2|\mathcal{A}) \le \mathbb{E}\left( \frac{C}{n_0^2} \sum_{i_1:Z_{i_1}=0} \epsilon_{i_1}^2 \middle| \mathcal{A} \right) \le \frac{C}{n_0}$$

The second term in $T_U$ can then be decomposed as

$$\frac{1}{n_0 n_1} \sum_{i_1:Z_{i_1}=0, i_2:Z_{i_2}=1} \mu_C^{\hat{\Phi}_D}(\vec{U}_{i_1}) K_h(\vec{U}_{i_1} - \vec{U}_{i_2}) - \theta_U$$

$$= \frac{1}{n_0 n_1} \sum_{i_1:Z_{i_1}=0, i_2:Z_{i_2}=1} \left( \mu_C^{\hat{\Phi}_D}(\vec{U}_{i_1}) K_h(\vec{U}_{i_1} - \vec{U}_{i_2}) - \int \mu_C^{\hat{\Phi}_D}(\vec{U}_{i_1}) K_h(\vec{U}_{i_1} - \vec{U}) f_T^{\hat{\Phi}_D}(\vec{U}) d\vec{U} \right)$$

$$+ \frac{1}{n_0} \sum_{i_1:Z_{i_1}=0} \left( \int \mu_C^{\hat{\Phi}_D}(\vec{U}_{i_1}) K_h(\vec{U}_{i_1} - \vec{U}) f_T^{\hat{\Phi}_D}(\vec{U}) d\vec{U} - \int \mu_C^{\hat{\Phi}_D}(\vec{U}) K_h(\vec{U} - \vec{U}') f_T^{\hat{\Phi}_D}(\vec{U}') d\vec{U} d\vec{U}' \right)$$

$$+ \int \mu_C^{\hat{\Phi}_D}(\vec{U}) K_h(\vec{U} - \vec{U}') f_T^{\hat{\Phi}_D}(\vec{U}') d\vec{U} d\vec{U}' - \int \mu_C^{\hat{\Phi}_D}(\vec{U}) f_T^{\hat{\Phi}_D}(\vec{U}) d\vec{U}$$

$$= A_3 + A_4 + A_5$$

Define

$$R(\vec{U}_{i_2}) = \frac{1}{n_0} \sum_{i_1:Z_{i_1}=0} \left( \mu_C^{\hat{\Phi}_D}(\vec{U}_{i_1}) K_h(\vec{U}_{i_1} - \vec{U}_{i_2}) - \int \mu_C^{\hat{\Phi}_D}(\vec{U}_{i_1}) K_h(\vec{U}_{i_1} - \vec{U}) f_T^{\hat{\Phi}_D}(\vec{U}) d\vec{U} \right).$$

Simple calculation yields

$$\mathbb{E}(R(\vec{U}_{i_2})) = 0 \qquad \text{and} \qquad \mathbb{E}(R^2(\vec{U}_{i_2})) \le C$$

Since $A_3 = \sum_{i_2:Z_{i_2}=1} R(\vec{U}_{i_2})/n_1$, we can have

$$\mathbb{E}(A_3^2|\mathcal{A}) \le \frac{C}{n_1}.$$

Because of property 1 in Proposition 1, we have

$$|A_4| \le \frac{C}{n_0^{\alpha_0/d}}.$$

In other words, we have

$$\mathbb{E}(A_4^2|\mathcal{A}) \leq \frac{C}{n_0^{2\alpha_0/d}}.$$

For the $A_5$, we apply the similar technique in the proof of Theorem 1

$$|A_5| \leq h^{\alpha+\beta}.$$

Putting $A_i, i = 1, \ldots, 5$ together suggests that

$$\mathbb{E}((T_U - \theta_U)^2|\mathcal{A}) \leq C \left( \frac{1}{n_0 n_1 h^d} + \frac{1}{n_0} + \frac{1}{n_1} + \frac{1}{n_0^{2\alpha_0/d}} + h^{2(\alpha+\beta)} \right).$$

**Step 2.** In this step, we show

$$\mathbb{E}((T_L - \theta_L)^2|\mathcal{A}) \leq C \left( \frac{1}{n_1} + \frac{1}{n_0^{2/d}} \right).$$

We write $T_L$ as

$$\begin{aligned}
T_L - \theta_L =& \frac{1}{n_0 n_1} \sum_{i_1:Z_{i_1}=0, i_2:Z_{i_2}=1} K_h(\vec{U}_{i_1} - \vec{U}_{i_2}) - \theta_L \\
=& \frac{1}{n_0 n_1} \sum_{i_1:Z_{i_1}=0, i_2:Z_{i_2}=1} K_h(\vec{U}_{i_1} - \vec{U}_{i_2}) - \int K_h(\vec{U}_{i_1} - \vec{U}) f_T^{\hat{\Phi}_D}(\vec{U}) d\vec{U} \\
&+ \frac{1}{n_0} \int K_h(\vec{U}_{i_1} - \vec{U}) f_T^{\hat{\Phi}_D}(\vec{U}) d\vec{U} - \int K_h(\vec{U}' - \vec{U}) f_T^{\hat{\Phi}_D}(\vec{U}) d\vec{U} d\vec{U}' \\
&+ \int K_h(\vec{U}' - \vec{U}) f_T^{\hat{\Phi}_D}(\vec{U}) d\vec{U} d\vec{U}' - \theta_L.
\end{aligned}$$

With the similar techniques in Step 1, we can conclude that

$$\mathbb{E}((T_L - \theta_L)^2|\mathcal{A}) \leq C \left( \frac{1}{n_1} + \frac{1}{n_0^{2/d}} \right).$$

**Step 3.** In the last step, we put the result of Steps 1 and 2 together, leading to the final conclusion. As $h$ is chosen as $h \to 0$ and $n^2 h^d \to 0$, we can know that

$$T_U \to_p \theta_U \qquad \text{and} \qquad T_L \to_p \theta_L.$$

Therefore, we can conclude that

$$\hat{\mu}_{CT} \to_p \mu_{CT}.$$

**Proof for projection density estimator** We omit the proof here, as we can apply the same parallel argument in the proof of Theorem 1.

