# OpenReview forum: "Treatment Effects Estimation By Uniform Transformer"
_ICLR.cc/2024/Conference — ICLR 2024 poster_

### Official Review · Reviewer_Ybv2 · 2023-10-31

**Soundness:** 3 good
**Presentation:** 3 good
**Contribution:** 3 good
**Rating:** 8
**Confidence:** 4

**Summary:**

The authors study the average treatment on the treated (ATT) in the Holder smooth paradigm of Robins et al. (2008), where minimax rates are known. Previous work characterizes the lower bound and shows that a higher order influence function estimator attains it. This paper proposes an alternative estimator based on kernel smoothing to also achieve the minimax lower bound. In the “hard cases” where the nuisances are non-smooth, relatively few estimators have been shown to have good properties; for the “easy cases” where the nuisances are collectively smooth enough, this is a very mature literature.

**Strengths:**

Originality: To my knowledge, this proposed estimator is new.

Quality: The results are cohesive. I recommend the authors mention Appendix B in the main text; I did not notice it at first.

Clarity: The writing is clear.

Significance: Proposing alternative estimators that achieve the minimax lower bound in hard cases is theoretically interesting, though perhaps not practically significant.

**Weaknesses:**

The introduction is too general for a paper that is ultimately about ATT only.

The kernel balancing weight literature is rich with many recent installments that should be at least mentioned in the introduction: Kallus (2020), Hirshberg et al. (2019), Singh (2021), and Bruns-Smith et al. (2023).

In Corollary 1, the convergence in distribution result for “easy” cases ends up being similar to several works that are unreferenced.

I will raise my score if these are addressed.

**Questions:**

What are the advantages of this approach versus the higher order influence function approach, which seems to have the same kinds of guarantees?

How do we choose the bandwidth in practice?

Is Theorem 2 a straightforward extension of the Robins et al. (2008) lower bound result? What are the aspects of it that are new? Close comparisons would help here.

I will raise my score if these are addressed.

---

> ### Author Response · Authors · 2023-11-13
> **Response to Reviewer Ybv2**
>
> We are very grateful for your careful reading of our work and your comments/suggestions,
> which helped improve the paper significantly. Here is our response for your comments.
>
> “I recommend the authors mention Appendix B in the main text; I did not notice it at first.”
>
> Response: Thanks for the suggestion. We now mention Appendix B at the end of Section 4.
>
> “The introduction is too general for a paper that is ultimately about ATT only.”
>
> Response: Thanks for the comment. Due to the page limit, we only present the estimator for average treatment on the treated (ATT). As illustrated in Appendix C.1, the method can also be used to estimate the average treatment effect (ATE). The theory can also be accordingly generalized straightforwardly.
>
> “The kernel balancing weight literature is rich with many recent installments that should be at least mentioned in the introduction: Kallus (2020), Hirshberg et al. (2019), Singh (2021), and Bruns-Smith et al. (2023).”
>
> Response: Thanks for the great suggestions. These references are added in the revised version of the paper.
>
> “In Corollary 1, the convergence in distribution result for “easy” cases ends up being similar to several works that are unreferenced.”
>
> Response: This is a great point. We have added some reference on $\sqrt{n}$-consistent estimator in the smooth case.
>
> “What are the advantages of this approach versus the higher order influence function approach, which seems to have the same kinds of guarantees?”
>
> Response: Thanks for the question. The bias reduction technique in the higher-order influence estimator is efficient but often involves elaborate expressions. In contrast, WUNT is a naive plug-in functional estimator after applying a uniform transformer. Because of the simple form, WUNT does not need an extra bias reduction process but only a carefully selected tuning parameter to reduce the bias. Such a naive plug-in functional estimator with a selected tuning parameter is also used to construct a simple estimator for the integrated squared density (Gine & Nickl, 2008).
>
> “How do we choose the bandwidth in practice?”
>
> Response: Thanks for the question. If we know the smoothness, we can use the bandwidth suggested in Theorem 1 and 3. If we do not know the smoothness, we can adopt Lepski’s method (Lepski, 1991; 1992) to select the tuning parameter in a data-driven way. We add the details to select bandwidth in Section C.3 in the appendix.
>
> “Is Theorem 2 a straightforward extension of the Robins et al. (2008) lower bound result? What are the aspects of it that are new? Close comparisons would help here.”
>
> Response: Thanks for the question. The setting in Robins et al. (2008, 2017) is a bit different from ours, although they are roughly equivalent. In Robins et al. (2017), the illustrative example is from missing data. If we translate it to a causal inference setting, $\alpha$ and $\beta$ are the smoothness levels of propensity score and response functions. However, our paper uses the smoothness levels of the probability density function and response functions. Because of different settings, the construction of the least favorable hypothesis is slightly different, so we write out the proof details for completeness. We also want to point out that the proof in our paper highly relies on the results of Robins et al. (2009).

---

### Official Review · Reviewer_424z · 2023-10-31

**Soundness:** 3 good
**Presentation:** 3 good
**Contribution:** 3 good
**Rating:** 6
**Confidence:** 4

**Summary:**

The paper introduces a method for estimating treatment effects using balancing weights, involving two key steps:
- a uniform transformation ensuring a uniform distribution density on the weight's denominator
- a density estimation of the transformed numerator

Under some regularity conditions and given that the density on the denominator is known, the authors establish an upper bound on the estimator's error rate and show that this rate of convergence is minimax. Additionally, they propose an adaptive uniform transformer that, when combined with the estimator, yields consistent treatment effect estimation.

**Strengths:**

- The authors introduce a novel effect estimation strategy utilizing balancing weights through uniform transformation, together with a new adaptive uniform transformer.

- The paper provides a sharp characterization (in terms of minimax optimality) of effect estimation through uniform transformers, which is new to the literature.

- The paper is clear and well-written.

**Weaknesses:**

- The use of a uniform transformer is not well-motivated. It is not clear to me, from the paper, why a uniform transformation is preferred over an estimator with densities $f_T(X)$ and $f_C(X)$ estimated separately without any transformation. Is it because it is harder to study such an estimator and thus harder to obtain the optimal bandwidth?

- The upper bound in Theorem 1 still relies largely on having accurate knowledge of $f_C$ (and also its smoothness level due to the $\alpha,\beta<\gamma$ constraint), and it doesn’t seem to me that the proposed method with the adaptive uniform transformer is able to achieve the minimax rate in general.

- The minimax optimality results are only for a very restricted class of problems.

Also see questions.

**Questions:**

- What is the difference between estimating both the densities of $f_T(X)$ and $f_C(X)$, as compared to the proposed strategy? In particular, under the conditions on $f_C(X)$ estimation posed in Corollary 1, is it possible for an estimator without the transformation to achieve the same rate?

- In general, what is the rate that can be achieved using the adaptive transformer? When adding strong enough regularity conditions, can it in general achieve the rate specified in Corollary 1? And what would those conditions be like?

- What is the advantage of using the proposed method, as compared to a conventional doubly robust estimator?

---

> ### Author Response · Authors · 2023-11-13
> **Response to Reviewer 424z (Part 1)**
>
> We thank the reviewer for the careful reading of our paper and constructive suggestions. Here is our response for your comments.
>
> “The use of a uniform transformer is not well-motivated. It is not clear to me, from the paper, why a uniform transformation is preferred over an estimator with densities $f_T(X)$ and $f_C(X)$ estimated separately without any transformation. Is it because it is harder to study such an estimator and thus harder to obtain the optimal bandwidth?”
>
> Response: This is a great question. Because of the uniform transformer, the estimator for $\mu_{CT}$ is just a simple U-statistics, so we can choose a suitable tuning parameter (which also relies on the smoothness level of the response function) to reduce bias. Whether this idea can extend if we estimate $f_T(X)$ and $f_C(X)$ separately is unclear. If we have an estimator for $f_C(X)$, a potential estimator is
> $$
> \hat{\mu}_{CT}={\sum_{i_1,i_2=1}^nY_{i_1}(1-Z_{i_1})\hat{f}_C^{-1}(X_{i_1})K_H(X_{i_1}- X_{i_2})Z_{i_2} \over \sum_{i_1,i_2=1}^n (1-Z_{i_1})\hat{f}_C^{-1}(X_{i_1})K_H(X_{i_1}- X_{i_2})Z_{i_2}}
> $$
> The theoretical analysis can be different. This direction deserves further study in the future.
>
> Another benefit of using a transformer is that Rosenblatt’s transformation relies only on the cumulative distribution function. In some applications, estimating the (conditional) cumulative distribution function is easier than the probability density function. For example, when $d=1$, the uniform transformer can be estimated by an empirical cumulative distribution function, and we do not need to estimate the probability density function.
>
> “The upper bound in Theorem 1 still relies largely on having accurate knowledge of $f_C(X)$ (and also its smoothness level due to the $\alpha, \beta<\gamma$ constraint), and it doesn’t seem to me that the proposed method with the adaptive uniform transformer is able to achieve the minimax rate in general.”
>
> Response: Thanks for raising this good point. The proof of Theorem 2 suggests that whether $f_C(X)$ is known in advance or not, the minimax rate of estimating $\mu_{CT}$ is the same. In addition, the existing minimax optimal estimators also need extra assumptions to achieve the minimax rate. For example, in Robins et al. (2017), Section 3.1 suggests that the higher order influence estimator also needs the function g to be known or easy to estimate (e.g., smooth enough). It is still an open question if the minimax optimal rate can be achieved without these extra assumptions. This is an interesting future direction.
>
> “The minimax optimality results are only for a very restricted class of problems.”
>
> Response: This is a valid point. In this paper, we only focus on the estimation of the average treatment effect and show the feasibility of a simple weighting estimator, demonstrating the potential of this idea. We will continue working in this direction and see whether we can extend the idea in simple U-statistics to more general problems.

---

> ### Author Response · Authors · 2023-11-13
> **Response to Reviewer 424z (Part 2)**
>
> “What is the difference between estimating both the densities of $f_T(X)$ and $f_C(X)$, as compared to the proposed strategy? In particular, under the conditions on $f_C(X)$ estimation posed in Corollary 1, is it possible for an estimator without the transformation to achieve the same rate?”
>
> Response: Thanks for the question. If we only focus on designing the best estimator for $f_T(X)$ and $f_C(X)$, then plugin them. This strategy could lead to a dominating bias, as argued in Robins et al. (2017). Therefore, when we design the estimator for $f_T(X)$ and $f_C(X)$, our target should be estimating the average treatment effect. It is not immediately clear whether the idea in the paper can be extended if we estimate $f_T(X)$ and $f_C(X)$ separately. If we have an estimator for $f_C(X)$, a potential estimator is
> $$
> \hat{\mu}_{CT}={\sum_{i_1,i_2=1}^nY_{i_1}(1-Z_{i_1})\hat{f}_C^{-1}(X_{i_1})K_H(X_{i_1}- X_{i_2})Z_{i_2}\over \sum_{i_1,i_2=1}^n (1-Z_{i_1})\hat{f}_C^{-1}(X_{i_1})K_H(X_{i_1}- X_{i_2})Z_{i_2}}
> $$
> The theoretical analysis can be different. This direction definitely deserves further study in the future.
>
>
> “In general, what is the rate that can be achieved using the adaptive transformer? When adding strong enough regularity conditions, can it in general achieve the rate specified in Corollary 1? And what would those conditions be like?”
>
> Response: This is a great question. The exact rate can be found at the end of step 1 in Proof of Theorem 3 (Section D.5). If the adaptive transformer is estimated from a data set with sample size in control $n_{0}$, the condition we need is
> $$
> n_0^{-2\alpha\over d}\le n_1^{-4(\alpha+\beta)\over d+2((\alpha+\beta)}.
> $$
> It is easier to satisfy when the dimension $d$ is smaller.
>
> “What is the advantage of using the proposed method, as compared to a conventional doubly robust estimator?”
>
> Response: Thanks for the great question. Section 5 of Robins et al. (2017) pointed out that conventional doubly robust estimators are essentially first-order estimators. When the underlying functions are non-smooth, Robins et al. (2017) demonstrates that the bias could dominate the estimation error and thus introduce a higher-order influence estimator to reduce bias better. Robins et al. (2017) shows that a higher-order influence estimator can effectively reduce bias to achieve minimax optimality when the underlying functions are non-smooth. Our estimator is also introduced to estimate the average treatment effect and achieve minimax optimality when the underlying functions are non-smooth. Different from a higher-order influence estimator, our proposed method is a weighting method. In a word, our estimator is a more efficient than the conventional doubly robust estimator in the non-smooth setting.

---

> > ### Comment · Reviewer_424z · 2023-11-22
> >
> > Thanks for addressing my comments.

---

### Official Review · Reviewer_oSbf · 2023-11-11

**Soundness:** 2 fair
**Presentation:** 2 fair
**Contribution:** 2 fair
**Rating:** 5
**Confidence:** 2

**Summary:**

This study proposes a novel framework for average treatment effects (ATEs) estimation by using uniform transformation. The authors develop an estimator that does not employ the exact estimation of the propensity score. Then, they show the finite-sample nonparametric minimax optimality for the estimator (but I think the estimator is not nonparametric in a usual sense...?) and asymptotic distribution.

**Strengths:**

Firstly, I note that I could not grasp the contributions of this study, especially the novelty of the proposal of the uniform transformer. Therefore, I could not assess this study well. If possible, I want to deepen my understanding and clarify the contributions of this study through an interaction with the authors.

The reasons why I could not understand the contributions are based on the following three points.

1. Necessity of the exact estimation of the propensity score.
In the 2000s, there were enormous arguments about what kind of weighting functions are effective in ATE estimation. As a result of long arguments, researchers found that using the **true** propensity score does not achieve the asymptotic lower bound proposed by Hahn (1998). That is, some finite-sample bias improves the asymptotic efficiency of estimators, as shown by Hirano et al. (2003). Does this study lie in those literature or completely discuss different topics?

2. Significance of the use of a kerne-based U-statistics.
To the best of my knowledge, existing studies such as Hirano et al. (2003) employ kernel-based U-statistics, at least in theoretical analysis. What is the difference from the existing approaches?

3. Minimax rate.
I could not understand why the authors use the minimax optimality for nonparametric estimators. The estimator used by the authors is typically referred to as a semiparametric estimator, which is characterized by nonparametric and parametric estimators. Then, the authors are interested in the estimator error of the parametric part of the semiparametric estimator. Therefore, I believe that the nonparametric minimax optimal rate is a bit meaningless in this context (in other words, I think that although the authors say that the proposed estimator is nonparametric, the estimator is (semi)parametric). Furthermore, the asymptotic variance in Corollary 1 is not efficient from the viewpoint of the semiparametric efficiency bound proposed by Hahn (1998). Is the estimator really efficient or minimax optimal?

Additionally, I could not understand connections to existing estimators whose asymptotic variance aligns with the semiparametric efficiency bound; that is, the authors' estimator cannot be more efficient than the existing efficient estimators. Overall, I could not understand the author's intent in this study.

My assessment may be biased my knowledge of existing literature of this topic, and I am not confident on my assessment. I hope that I can deepen the understanding via communication with the authors.

**Weaknesses:**

See above.

**Questions:**

See above.

**Details Of Ethics Concerns:**

None.

---

> ### Author Response · Authors · 2023-11-13
> **Response to Reviewer oSbf (Part 1)**
>
> Thank you very much for your detailed and constructive comments. Here is our response for your comments.
>
> “Necessity of the exact estimation of the propensity score. In the 2000s, there were enormous arguments about what kind of weighting functions are effective in ATE estimation. As a result of long arguments, researchers found that using the true propensity score does not achieve the asymptotic lower bound proposed by Hahn (1998). That is, some finite-sample bias improves the asymptotic efficiency of estimators, as shown by Hirano et al. (2003). Does this study lie in those literature or completely discuss different topics?”
>
> Response: Thanks for the great question. Similar to Robins et al. (2008, 2017), the discussion in this paper assumes we don’t know the true propensity score. Our main point is that the best way to estimate weights or propensity scores does not necessarily lead to the best estimator for the average treatment effect when the probability density functions and response functions are not smooth. In the non-smooth setting, we need to carefully balance the bias and variance in the final estimator for the average treatment effect.
>
> “Significance of the use of a kerne-based U-statistics. To the best of my knowledge, existing studies such as Hirano et al. (2003) employ kernel-based U-statistics, at least in theoretical analysis. What is the difference from the existing approaches?”
>
> Response: Thanks for the question. In Hirano et al. (2003), the propensity score is estimated by a Series Logit Estimator (SLE), which is then used in the weighting method. Most previous literature focuses on a setting where probability density functions/propensity score and response functions are smooth enough. For example, Hirano et al. (2003) assumes the propensity score is continuously differentiable of order $s\ge 7d$ (Assumption 4). In Hahn (1998), propensity score and response functions are assumed to be continuously differentiable of all orders (Assumption (iv) in Theorem 6). Chan et al. (2016) assumes that the propensity score is $s$-times continuously differentiable, where $s > 13d$, and the response function is $t$-times continuously differentiable, where $t> 3d/s$. Our paper's discussion mainly focuses on a setting where the sum of the smoothness levels of probability density and response functions is smaller than $d/2$. Whether these methods can still work effectively in the non-smooth setting is unclear. Besides our proposed method, another estimator designed for a non-smooth setting is the higher-order influence estimator introduced by Robins et al. (2017). The bias reduction technique in the higher-order influence estimator involves elaborate expressions. In contrast, our proposed method is a simple functional estimator after applying a uniform transformer. Because of the simple form, the new method does not need an extra bias reduction process but only a carefully selected tuning parameter to reduce the bias.

---

> ### Author Response · Authors · 2023-11-13
> **Response to Reviewer oSbf (Part 2)**
>
> “Minimax rate. I could not understand why the authors use the minimax optimality for nonparametric estimators. The estimator used by the authors is typically referred to as a semiparametric estimator, which is characterized by nonparametric and parametric estimators. Then, the authors are interested in the estimator error of the parametric part of the semiparametric estimator. Therefore, I believe that the nonparametric minimax optimal rate is a bit meaningless in this context (in other words, I think that although the authors say that the proposed estimator is nonparametric, the estimator is (semi)parametric). Furthermore, the asymptotic variance in Corollary 1 is not efficient from the viewpoint of the semiparametric efficiency bound proposed by Hahn (1998). Is the estimator really efficient or minimax optimal?”
>
> Response: Thanks for the question. As we mentioned before, there are two regimes in estimating the average treatment effect: $\alpha+\beta>d/2$ (smooth) and $\alpha+\beta<d/2$ (non-smooth). In the smooth regime, $\sqrt{n}$-consistent estimators exist (i.e., the convergence rate is $O(1/\sqrt{n})$), and there is a very mature literature. For example, Hirano et al. (2003) and Hahn (1998) lie in this regime (the estimators in these two papers need stronger conditions than $\alpha+\beta>d/2$). To compare different $\sqrt{n}$-consistent estimators, we can compare their variance from the viewpoint of the semiparametric efficiency. The semiparametric efficiency bound is an asymptotic lower bound on variances of estimators, so a method that achieves such a lower bound can be called an optimal method. When it comes to the non-smooth regime, $\sqrt{n}$-consistent estimators do not exist anymore, and the convergence rate of estimators is usually slower than $O(1/\sqrt{n})$. Minimax optimality is usually used to characterize the optimality of a method in the non-smooth regime (see Robins et al. (2017)). Compared with the smooth regime, relatively few estimators demonstrate good properties in the non-smooth regime. In this paper, we focus on the non-smooth regime, introduce an estimator designed for the non-smooth regime, and show it can achieve a minimax optimal rate in the non-smooth regime.
>
> “Additionally, I could not understand connections to existing estimators whose asymptotic variance aligns with the semiparametric efficiency bound; that is, the authors' estimator cannot be more efficient than the existing efficient estimators. Overall, I could not understand the author's intent in this study.”
>
> Response: Thanks for the comments. Again, this paper focuses on a non-smooth setting, while most existing methods are designed for a smooth setting. It is unclear if these methods designed for smooth settings are still efficient in non-smooth settings. This paper aims to develop a method that can work well in a non-smooth setting.

---

### Meta-Review · Area_Chair_x7Y5 · 2023-12-12

**Metareview:**

The authors develop a method for causal effect estimation in the non-smooth setting based on weighting to directly target treatment effect estimation. The approach is called the uniform transformer. The authors study and develop the theoretical properties of this new estimator and have some numerical experiments in the appendix. All of the reviewers were positive on this paper and overall it's a nice theoretical contribution in causal inference.

**Justification For Why Not Higher Score:**

The paper makes a focused theoretical contribution.

**Justification For Why Not Lower Score:**

The theoretical contribution addresses a problem that is active under study.

---

### Decision · Program_Chairs · 2024-01-16

Accept (poster)